# UNDERSTANDING OVERFITTING IN REWEIGHTING ALGORITHMS FOR WORST-GROUP PERFORMANCE

## ABSTRACT

Prior work has proposed various reweighting algorithms to improve the worst-group performance of machine learning models for fairness. However, Sagawa et al. (2020a) empirically found that these algorithms overfit easily in practice under the overparameterized setting, where the number of model parameters is much greater than the number of samples. In this work, we provide a theoretical backing to the empirical results above, and prove the pessimistic result that reweighting algorithms always overfit. Specifically we prove that with reweighting, an overparameterized model always converges to the *same ERM interpolator* that fits all training samples, and consequently its worst-group test performance will drop to the same level as ERM in the long run. That is, we cannot hope for reweighting algorithms to converge to a *different interpolator* than ERM with potentially better worst-group performance. Then, we analyze whether adding regularization helps fix the issue, and we prove that for regularization to work, it must be large enough to prevent the model from achieving small training error. Our results suggest that large regularization (or early stopping) and data augmentation are necessary for reweighting algorithms to achieve high worst-group test performance.

## 1 INTRODUCTION

It has been well established by prior work that overparameterized models, whose number of parameters is much larger than the number of training samples, can empirically achieve high test performance on a variety of tasks, in contrast to the theory that models with too many parameters could have large generalization error.

This high performance however is *on average*; a large body of prior work (Hovy & Søgaard, 2015; Blodgett et al., 2016; Tatman, 2017) showed that these models tend to learn *spurious features*, such as learning the background in image classification instead of the object, and learning keywords like "not" in language sentiment analysis instead of really understanding the sentences. Consequently, these models are *unfair*, i.e. they fail on certain minority groups (such as positive sentences containing "not") while still having high average-case performance. To solve this problem, people have proposed various *reweighting algorithms* to improve the model's worst-group performance, such as upweighting the minority groups or using distributionally robust optimization (DRO) based methods (Shimodaira, 2000; Hashimoto et al., 2018; Duchi & Namkoong, 2018; Sagawa et al., 2020a).

While reweighting algorithms in principle can improve the worst-group performance compared to vanilla empirical risk minimization (ERM), previous work empirically found that when applied to modern overparameterized models, these methods could overfit very easily, so that they have poor test worst-group performance. For example, Sagawa et al. (2020a) studied a reweighting algorithm called group DRO. They found that compared to ERM, group DRO does improve the worst-group test accuracy by a large margin at the early stage of training. However, if no regularization is applied, then as training goes on, the worst-group test accuracy of group DRO will drop significantly and eventually to a level almost the same as ERM. Some previous work tried to explain why reweighting algorithms can overfit so easily. For instance, Sagawa et al. (2020b) argued that with these algorithms, an overparameterized model would typically memorize all training samples in the minority groups while still learning the spurious features from the majority groups.

In this work, we aim to understand the overfitting phenomenon in reweighting algorithms by studying their implicit biases. Specifically, we prove for a family of overparameterized neural networks

that for almost all reweighting algorithms, the model always converges to the *same interpolator* that fits all training samples, no matter the reweighting. Since ERM is a special case of such reweighting algorithms (where each sample receives the same weight), this means that the implicit biases of all reweighting algorithms are equivalent to that of ERM. Consequently, the model trained by any reweighting algorithm always overfits to the ERM interpolator, so we cannot hope for its worst-group test performance to be better than ERM. In short, *reweighting algorithms always overfit*.

Given this pessimistic result, we analyze whether regularization can help mitigate overfitting, as proposed by Sagawa et al. (2020a). We find that a necessary condition for regularization to work is that it considerably lowers the training performance. Specifically, we prove that if the overparameterized model trained by a reweighting algorithm with regularization can still perform almost perfectly on the training set, then overfitting is still inevitable. This explains why in practice we need very large regularization that prevents the model from achieving nearly zero training error to avoid overfitting.

Our results have two important consequences for practice: (i) We should always use large regularization or early stopping when optimizing for worst-group performance; (ii) We should always try to obtain more training samples, e.g. with strong data augmentation or semi-supervised learning.

## 1.1 RELATED WORK

**Group fairness.** Group fairness in machine learning was first studied in Hardt et al. (2016) and Zafar et al. (2017), where they required the model to perform equally well over all groups. Later, Hashimoto et al. (2018) studied another type of group fairness called Rawlsian max-min fairness (Rawls, 2001), which does not require equal performance but rather requires high performance on the worst-off group. The problem we study in this paper is most closely related to Rawlsian max-min fairness. A large body of recent work in machine learning have studied how to improve this worst-group performance (Duchi & Namkoong, 2018; Oren et al., 2019; Xu et al., 2020; Liu et al., 2021; Zhai et al., 2021). Recent work however observe that these approaches, when used with modern overparameterized models, easily overfit (Sagawa et al., 2020a;b). Apart from group fairness, there are also other notions of fairness, such as individual fairness (Dwork et al., 2012; Zemel et al., 2013) and counterfactual fairness (Kusner et al., 2017), which we do not study in this work.

**Implicit bias under the overparameterized setting.** For overparameterized models, there could be many model parameters which all minimize the training loss. In such cases, it is of interest to study the implicit bias of specific optimization algorithms such as gradient descent i.e. to what training loss minimizer the model parameters will converge to (Du et al., 2019; Allen-Zhu et al., 2019). Our results use the NTK formulation of wide neural networks (Jacot et al., 2018), and specifically we use linearized neural networks to approximate such wide neural networks following Lee et al. (2019). There is some criticism of this line of work, e.g. Chizat et al. (2019) argued that infinitely wide neural networks fall in the "lazy training" regime and results might not be transferable to general neural networks. Nonetheless such wide neural networks are being widely studied in recent years, since they provide considerable insights into the behavior of more general neural networks, which are typically intractable to analyze otherwise.

## 2 PRELIMINARIES

Consider a data domain $\mathcal{X} \times \mathcal{Y} \subseteq \mathbb{R}^d \times \mathbb{R}$ that consists of $K$ groups (subdomains)[1], where each data point belongs to one of the groups[2]. We assume that the input space $\mathcal{X}$ is a subset of the unit ball of $\mathbb{R}^d$, such that any $\boldsymbol{x} \in \mathcal{X}$ satisfies $\|\boldsymbol{x}\|_2 \leq 1$. We are given a training set $\{(\boldsymbol{x}_i, y_i)\}_{i=1}^n$ *i.i.d.* sampled from some underlying distribution $P$ over $\mathcal{X} \times \mathcal{Y}$. Let the $K$ groups be $\mathcal{D}_1, \cdots, \mathcal{D}_K$ where each $\mathcal{D}_i$ is a subset of $\mathcal{X} \times \mathcal{Y}$. Let $P_k(\boldsymbol{z}) = P(\boldsymbol{z}|\boldsymbol{z} \in \mathcal{D}_k)$ be the conditional data distribution over $\mathcal{D}_k$, where $\boldsymbol{z} = (\boldsymbol{x}, y)$. Denote $\boldsymbol{X} = (\boldsymbol{x}_1, \cdots, \boldsymbol{x}_n) \in \mathbb{R}^{d \times n}$, and $\boldsymbol{Y} = (y_1, \cdots, y_n) \in \mathbb{R}^n$; for any function $g : \mathcal{X} \mapsto \mathbb{R}$, we overload notation and use $g(\boldsymbol{X}) = (g(\boldsymbol{x}_1), \cdots, g(\boldsymbol{x}_n))$. Let the loss function be $\ell : \mathcal{Y} \times \mathcal{Y} \to [0, 1]$. In vanilla training, the goal is to minimize the expected risk denoted by $\mathcal{R}(f; P) = \mathbb{E}_{\boldsymbol{z} \sim P}[\ell(f(\boldsymbol{x}), y)]$, which is done by minimizing the empirical risk $\hat{\mathcal{R}}(f) = \frac{1}{n} \sum_{i=1}^n \ell(f(\boldsymbol{x}_i), y_i)$.

---

[1]We prove our results for $\mathcal{Y} \subseteq \mathbb{R}$, but our results can be easily extended to the multi-class scenario $\mathcal{Y} \subseteq \mathbb{R}^m$.

[2]This is the non-overlapping setting. There is also the overlapping setting where groups can overlap with each other. We focus on the non-overlapping setting in this paper.

For tasks requiring high worst-group performance, the goal is to train a model $f : \mathcal{X} \to \mathcal{Y}$ that performs well over every $P_k$, which can be achieved by minimizing the *worst-group risk* defined as

$$\mathcal{R}_{\max}(f; P) = \max_{k=1,\cdots,K} \mathcal{R}(f; P_k) = \max_{k=1,\cdots,K} \mathbb{E}_{\boldsymbol{z} \sim P}[\ell(f(\boldsymbol{x}), y) | z \in \mathcal{D}_k] \qquad (1)$$

### 2.1 REWEIGHTING ALGORITHMS

Most existing methods that minimize the worst-group risk are *reweighting algorithms* that assign each sample with a weight during training and minimize the weighted average risk. At time $t$, we assign a weight $q_i^{(t)}$ to sample $\boldsymbol{z}_i$, and minimize the weighted empirical risk:

$$\hat{\mathcal{R}}_{\boldsymbol{q}^{(t)}}(f) = \sum_{i=1}^n q_i^{(t)} \ell(f(\boldsymbol{x}_i), y_i) \qquad (2)$$

where $\boldsymbol{q}^{(t)} = (q_1^{(t)}, \cdots, q_n^{(t)})$ and $q_1^{(t)} + \cdots + q_n^{(t)} = 1$.

A *static reweighting algorithm* assigns to each $\boldsymbol{z}_i = (\boldsymbol{x}_i, y_i)$ a fixed weight $q_i$ that does not change during training, i.e. $q_i^{(t)} \equiv q_i$. A famous example is *Importance Weighting* (IW, Shimodaira (2000)), in which if $\boldsymbol{z}_i \in \mathcal{D}_k$ and the size of $\mathcal{D}_k$ is $n_k$, then $q_i = (Kn_k)^{-1}$. Under IW, each group has the same weight, and the reweighted empirical risk is a simple (unweighted) average of the empirical risk over each group, so that each group has an equal contribution to the overall risk objective. Note that ERM is also a special case of static reweighting algorithms: by assigning $q_1 = \cdots = q_n = 1/n$.

On the other hand, in a *dynamic reweighting algorithm*, $\boldsymbol{q}^{(t)}$ changes with $t$. Specifically, it up-weights samples over which the model has a high risk in order to help the model learn "hard" samples. A popular dynamic reweighting algorithm is *Group DRO* (Sagawa et al., 2020a). Denote the empirical risk over group $k$ by $\hat{\mathcal{R}}_k(f)$, and the model at time $t$ by $f^{(t)}$. Group DRO sets $q_i^{(t)} = g_k^{(t)}/n_k$ for all $\boldsymbol{z}_i \in \mathcal{D}_k$ where $g_k^{(t)}$ is the group weight that is updated by

$$g_k^{(t)} \propto g_k^{(t-1)} \exp\left(\nu \hat{\mathcal{R}}_k(f^{(t-1)})\right) \qquad (\forall k = 1, \cdots, K) \qquad (3)$$

for some $\nu > 0$, and then normalized so that $q_1^{(t)} + \cdots + q_n^{(t)} = 1$. Sagawa et al. (2020a) proved a convergence rate theorem (their Proposition 2) showing that in the convex setting, the worst-group training risk of Group DRO converges to the global minimum with the rate $O(t^{-1/2})$.

There are many other reweighting algorithms. Particularly, all variants of DRO and DRO-based methods like CVaR and $\chi^2$-DRO are reweighting algorithms. See Appendix A for more examples.

### 2.2 REWEIGHTING ALGORITHMS CAN EASILY OVERFIT

In this section, we will empirically demonstrate that while IW and Group DRO can achieve higher worst-group test performances than ERM at the early stage of training, they can easily overfit after a number of training epochs.

Following Sagawa et al. (2020a), we conduct the experiment on two datasets: Waterbirds and CelebA. Each dataset contains a binary confounding variable $a$ and a binary target variable $y$, dividing the dataset into four groups (four combinations of $(a, y)$). In Waterbirds $y$ is the type of the bird and $a$ is the background; In CelebA $y$ is whether the person has blond hair and $a$ is whether the person is male. On each dataset, a model trained by ERM always exhibits a very strong empirical correlation between $y$ and $a$, so its performance on one of the groups is extremely poor. The goal is to make the model perform well on every group. See Appendix C.1 of Sagawa et al. (2020a) for detailed information of these datasets.

On each dataset, we use the ResNet18 model as the classifier and optimize it with momentum SGD. We run each of the three algorithms: ERM, IW and group DRO (GDRO), for 500 epochs on Waterbirds and 200 epochs on CelebA, and plot the average training/test and worst-group (WG) training/test accuracy curves throughout training in Figure 1. From the plots we can conclude that:

- All algorithms can achieve and maintain high average training/test accuracy throughout training, i.e. there is almost no overfitting in the average test accuracy.

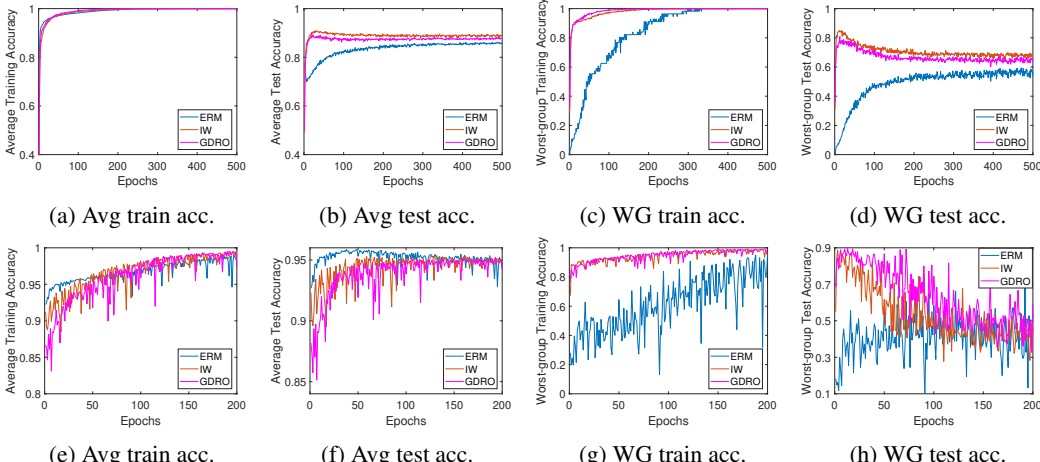

| (a) Avg train acc. | (b) Avg test acc. | (c) WG train acc. | (d) WG test acc. |
| (e) Avg train acc. | (f) Avg test acc. | (g) WG train acc. | (h) WG test acc. |

Figure 1: Performances of ERM, IW and Group DRO. First row: Waterbirds. Second row: CelebA.

- Regarding the worst-group test accuracy, while the two reweighting algorithms outperform ERM by a large margin at the early epochs, they overfit very quickly. On CelebA after roughly 100 epochs, the worst-group test accuracies of the two reweighting algorithms become the same as ERM. On Waterbirds, the worst-group test performances of IW and Group DRO drop significantly after around 30 epochs though they are still better than ERM.

## 3 IMPLICIT BIASES OF REWEIGHTING ALGORITHMS

In the previous section, we empirically demonstrated that the worst-group test performances of reweighting algorithms converge to the same level as ERM. To theoretically understand why this happens in practice, we analyze the implicit biases of reweighting algorithms. Our main theorem (Theorem 6) states that almost all reweighting algorithms (including ERM) have equivalent implicit biases, in the sense that they converge to the same interpolator. Meanwhile, it is observed in practice that the ERM interpolator has a poor worst-group test performance. This leads to the pessimistic result that *reweighting algorithms always overfit*. All proofs can be found in Appendix B.

### 3.1 LINEAR MODELS

We first demonstrate this pessimistic result on simple linear models to provide our readers with a key intuition, and later we will apply this same intuition to neural networks. Let the linear model be $f(\boldsymbol{x}) = \langle \theta, \boldsymbol{x} \rangle$, where $\theta \in \mathbb{R}^d$. In the overparameterized setting, we have $d > n$. Consider using the squared loss $\ell(\hat{y}, y) = \frac{1}{2}(\hat{y} - y)^2$, and minimizing the weighted empirical risk with gradient descent:

$$\theta^{(t+1)} = \theta^{(t)} - \eta \sum_{i=1}^{n} q_i^{(t)} \nabla_\theta \ell(f^{(t)}(\boldsymbol{x}_i), y_i) \qquad (4)$$

where $\eta > 0$ is the learning rate. For a linear model with the squared loss, the update rule is

$$\theta^{(t+1)} = \theta^{(t)} - \eta \sum_{i=1}^{n} q_i^{(t)} \boldsymbol{x}_i (f^{(t)}(\boldsymbol{x}_i) - y_i) \qquad (5)$$

It is a well known result that under the overparameterization setting where $d > n$, if $\boldsymbol{x}_1, \cdots, \boldsymbol{x}_n$ are linearly independent, then with a sufficiently small $\eta$, a linear model trained by ERM can always converge to an *interpolator* which fits all training samples (i.e. $\theta^{(t)} \to \theta^*$ such that $\langle \theta^*, \boldsymbol{x}_i \rangle = y_i$ for all $i$). Here the linear independence is necessary, because otherwise in the extreme case where $\boldsymbol{x}_1 = \boldsymbol{x}_2$ but $y_1 \neq y_2$, the model cannot fit $(\boldsymbol{x}_1, y_1)$ and $(\boldsymbol{x}_2, y_2)$ simultaneously.

In this section, we aim to extend this ERM convergence analysis to general reweighting algorithms. Our results require the following assumption:

**Assumption 1.** *There exist constants* $q_1, \cdots, q_n$ *such that for all* $i$, $q_i^{(t)} \to q_i$ *as* $t \to \infty$. *And* $\min_i q_i = q^* > 0$.

This assumption avoids the scenario where there is some $i$ such that $q_i^{(t)} \approx 0$ for all $t$, in which case the model could never fit $\boldsymbol{z}_i$. Assumption 1 empirically holds for Group DRO on Waterbirds and CelebA (see Appendix D.2). Under this assumption, we can prove that the model always converges to an interpolator:

**Theorem 1.** *For any reweighting algorithm satisfying Assumption 1, if $\boldsymbol{x}_1, \cdots, \boldsymbol{x}_n$ are linearly independent, then there exists an $\eta_0 > 0$ such that for any $\eta \leq \eta_0$, as $t \to \infty$, $\theta^{(t)}$ converges to some interpolator $\theta^*$ such that for all $i$, $\langle \theta^*, \boldsymbol{x}_i \rangle = y_i$.*

We now make the following key observation regarding the update rule (5): $\theta^{(t+1)} - \theta^{(t)}$ is a linear combination of $\boldsymbol{x}_1, \cdots, \boldsymbol{x}_n$ for all $t$, and thus $\theta^{(t)} - \theta^{(0)}$ always lies in the linear subspace $\mathrm{span}(\boldsymbol{x}_1, \cdots, \boldsymbol{x}_n)$. Note that this is an $n$-dimensional linear subspace if $\boldsymbol{x}_1, \cdots, \boldsymbol{x}_n$ are linearly independent, and by Cramer's rule, there is exactly one $\tilde{\theta}$ in this subspace such that $\langle \tilde{\theta} + \theta^{(0)}, \boldsymbol{x}_i \rangle = y_i$ for all $i$, which implies that $\theta^* = \tilde{\theta} + \theta^{(0)}$ is unique. Together with Theorem 1, this leads to:

**Theorem 2.** *If $\boldsymbol{x}_1, \cdots, \boldsymbol{x}_n$ are linearly independent, then there exists $\eta_0 > 0$ such that for any reweighting algorithm satisfying Assumption 1, and any $\eta \leq \eta_0$, $\theta^{(t)}$ converges to the same interpolator $\theta^*$ that does not depend on $q_i^{(t)}$.*

Note that ERM is also a reweighting algorithm satisfying Assumption 1. Therefore, we have essentially proved the following result: *The implicit bias of any reweighting algorithm satisfying Assumption 1 is equivalent to ERM, so reweighting algorithms always overfit*[3].

The key intuition here is that no matter what reweighting algorithm we use, $\theta^{(t)} - \theta^{(0)}$ always lies in a low-dimensional subspace, in which the interpolator is unique. Therefore, as long as a model trained by the algorithm converges to some interpolator, it must converge to that unique interpolator, which means that the implicit bias of the algorithm is equivalent to ERM.

### 3.2 Linearized Neural Networks

Now we prove the same result for neural networks. Of course it would be very hard to prove it for all neural networks. However, we can prove the result for a family of overparameterized neural networks that can be approximated by their linearized counterparts Lee et al. (2019). Denote the neural network at time $t$ by $f^{(t)}(\boldsymbol{x}) = f(\boldsymbol{x}; \theta^{(t)})$ which is parameterized by $\theta^{(t)} \in \mathbb{R}^p$ where $p$ is the number of parameters. The *linearized neural network* of $f^{(t)}(\boldsymbol{x})$ is defined as

$$f_{\mathrm{lin}}^{(t)}(\boldsymbol{x}) = f^{(0)}(\boldsymbol{x}) + \langle \theta^{(t)} - \theta^{(0)}, \nabla_\theta f^{(0)}(\boldsymbol{x}) \rangle \tag{6}$$

where we use the shorthand $\nabla_\theta f^{(0)}(\boldsymbol{x}) := \nabla_\theta f(\boldsymbol{x}; \theta)\big|_{\theta=\theta_0}$. Consider training $f_{\mathrm{lin}}^{(t)}(\boldsymbol{x})$ via gradient descent on the reweighted risk (as in (4)) using the squared loss. Given a training set $\{(\boldsymbol{x}_i, y_i)\}_{i=1}^n$, we can construct a new training set $\{(\nabla_\theta f^{(0)}(\boldsymbol{x}_i), y_i - f^{(0)}(\boldsymbol{x}_i))\}_{i=1}^n$, so that training a linearized neural network on the original training set is equivalent to training a linear model on the new training set. Based on this observation, we have the following corollary of Theorem 2:

**Corollary 3.** *If $\nabla_\theta f^{(0)}(\boldsymbol{x}_1), \cdots, \nabla_\theta f^{(0)}(\boldsymbol{x}_n)$ are linearly independent, then there exists $\eta_0 > 0$ such that for any reweighting algorithm satisfying Assumption 1, and any $\eta \leq \eta_0$, $\theta^{(t)}$ converges to the same interpolator $\theta^*$ that does not depend on $q_i$.*

Here we are still using the key intuition: $\theta^{(t)} - \theta^{(0)}$ always lies in the $n$-dimensional linear subspace $\mathrm{span}\left(\nabla_\theta f^{(0)}(\boldsymbol{x}_1), \cdots, \nabla_\theta f^{(0)}(\boldsymbol{x}_n)\right)$. By Cramer's rule, there is a unique interpolator $\theta^*$ such that $\theta^* - \theta^{(0)} \in \mathrm{span}\left(\nabla_\theta f^{(0)}(\boldsymbol{x}_1), \cdots, \nabla_\theta f^{(0)}(\boldsymbol{x}_n)\right)$, and $\theta^{(t)}$ always converges to that $\theta^*$. Thus, we have essentially proved that for linearized neural networks, reweighting algorithms always overfit.

Now let us delve deeper into the training dynamics of a linearized neural network. Note that $\nabla_\theta f_{\mathrm{lin}}^{(t)}(\boldsymbol{X}) = \nabla_\theta f^{(0)}(\boldsymbol{X}) \in \mathbb{R}^{p \times n}$, so the change in the training function value vector is

$$f_{\mathrm{lin}}^{(t+1)}(\boldsymbol{X}) - f_{\mathrm{lin}}^{(t)}(\boldsymbol{X}) = -\eta \, \nabla_\theta f^{(0)}(\boldsymbol{X})^\top \nabla_\theta f^{(0)}(\boldsymbol{X}) \boldsymbol{Q}^{(t)} \, \nabla_{\hat{y}} \ell(f_{\mathrm{lin}}^{(t)}(\boldsymbol{X}), \boldsymbol{Y}) \tag{7}$$

---

[3]By *overfit*, we are saying that the training error of the model trained by the reweighting algorithm will converge to zero, but the worst-group test performance will converge to the same low level as ERM.

where $\boldsymbol{Q}^{(t)} = \mathrm{diag}(q_1^{(t)}, \cdots, q_n^{(t)})$. The function value vector moves along the kernel gradient with respect to $\Theta_{\boldsymbol{q}^{(t)}}^{(0)} = \nabla_\theta f^{(0)}(\boldsymbol{X})^\top \nabla_\theta f^{(0)}(\boldsymbol{X}) \boldsymbol{Q}^{(t)}$. Meanwhile, the *neural tangent kernel* (NTK, Jacot et al. (2018)) is $\Theta^{(0)}(\boldsymbol{x}, \boldsymbol{x}') = \nabla_\theta f^{(0)}(\boldsymbol{x})^\top \nabla_\theta f^{(0)}(\boldsymbol{x}')$ , and the *Gram matrix* is $\Theta^{(0)} = \Theta^{(0)}(\boldsymbol{X}, \boldsymbol{X})$, so $\Theta_{\boldsymbol{q}^{(t)}}^{(0)} = \Theta^{(0)} \boldsymbol{Q}^{(t)}$. We can thus extend our result for gradient descent on linearized neural networks to a kernel gradient descent algorithm as above.

### 3.3 Wide fully-connected neural networks

Now we prove the result for sufficiently wide fully-connected neural networks, which can be approximated by the linearized neural networks. First we define a fully-connected neural network with $L$ hidden layers (we always assume $L \geq 1$ so there is at least one hidden layer). Let $\boldsymbol{h}^l$ and $\boldsymbol{x}^l$ be the pre- and post-activation outputs of layer $l$, and $d_l$ be the width of layer $l$. Let $\boldsymbol{x}^0 = \boldsymbol{x}$ and $d_0 = d$. Define the neural network as

$$\begin{cases} \boldsymbol{h}^{l+1} = \dfrac{W^l}{\sqrt{d_l}} \boldsymbol{x}^l + \beta \boldsymbol{b}^l \\ \boldsymbol{x}^{l+1} = \sigma(\boldsymbol{h}^{l+1}) \end{cases} \qquad (l = 0, \cdots, L) \tag{8}$$

where $\sigma$ is a non-linear activation function, $W^l \in \mathbb{R}^{d_{l+1} \times d_l}$ and $W^L \in \mathbb{R}^{1 \times d_L}$. The parameters $\theta$ consist of $W^0, \cdots, W^L$ and $b^0, \cdots, b^L$ ($\theta$ is the concatenation of all flattened weights and biases). The final output of the neural network is $f(\boldsymbol{x}) = \boldsymbol{h}^{L+1}$. And let the neural network be initialized as

$$\begin{cases} W_{i,j}^{l(0)} \sim \mathcal{N}(0, 1) \\ \boldsymbol{b}_j^{l(0)} \sim \mathcal{N}(0, 1) \end{cases} \quad (l = 0, \cdots, L-1) \qquad \text{and} \qquad \begin{cases} W_{i,j}^{L(0)} = 0 \\ \boldsymbol{b}_j^{L(0)} \sim \mathcal{N}(0, 1) \end{cases} \tag{9}$$

We also need the following assumption for our approximation theorem:

**Assumption 2.** *$\sigma$ is differentiable everywhere, and both $\sigma$ and $\dot{\sigma}$ are Lipschitz.*[4]

**Difference from Jacot et al. (2018).** Our initialization (9) is different from the original one in Jacot et al. (2018) in the last (output) layer. For the output layer, we use the zero initialization $W_{i,j}^{L(0)} = 0$ instead of the Gaussian initialization $W_{i,j}^{L(0)} \sim \mathcal{N}(0, 1)$. This modification enables us to accurately approximate the neural network with its linearized counterpart (6), as we notice that the proofs in Lee et al. (2019) (particularly the proofs of their Theorem 2.1 and their Lemma 1 in Appendix G) are flawed. In Appendix C we will explain what goes wrong in their proofs and how we manage to fix the proofs with our modification.

For our new initialization, we still have the following NTK theorem:

**Theorem 4.** *If $\sigma$ is Lipschitz and $d_l \to \infty$ for $l = 1, \cdots, L$ sequentially, then $\Theta^{(0)}(\boldsymbol{x}, \boldsymbol{x}')$ converges in probability to a non-degenerated[5] deterministic limiting kernel $\Theta(\boldsymbol{x}, \boldsymbol{x}')$.*

The kernel Gram matrix $\Theta = \Theta(\boldsymbol{X}, \boldsymbol{X}) \in \mathbb{R}^{n \times n}$ is a positive semi-definite symmetric matrix. Denote its largest and smallest eigenvalues by $\lambda^{\mathrm{max}}$ and $\lambda^{\mathrm{min}}$. Note that $\Theta$ is non-degenerated, so we assume that $\lambda^{\mathrm{min}} > 0$ (which holds almost surely in the overparameterized setting where $d_L \gg n$). Then, we can prove the following approximation theorem:

**Theorem 5** (Approximation Theorem). *Let $\eta^* = (\lambda^{\mathrm{min}} + \lambda^{\mathrm{max}})^{-1}$. For a fully-connected neural network $f^{(t)}$ that satisfies Assumption 2 and is trained by any reweighting algorithm satisfying Assumption 1, let $f_{\mathrm{lin}}^{(t)}$ be its linearized neural network which is trained by the same reweighting algorithm (i.e. $\forall i, t$, $q_i^{(t)}$ are the same for both networks). If $d_1 = d_2 = \cdots = d_L = \tilde{d}$ and $\lambda^{\mathrm{min}} > 0$, then for any $\delta > 0$, there exists $\tilde{D} > 0$ and a constant $C$ such that as long as $\eta \leq \eta^*$ and $\tilde{d} \geq \tilde{D}$, for any test point $\boldsymbol{x} \in \mathbb{R}^d$ such that $\|\boldsymbol{x}\|_2 \leq 1$, with probability at least $1 - \delta$ over random initialization,*

$$\sup_{t \geq 0} \left| f_{\mathrm{lin}}^{(t)}(\boldsymbol{x}) - f^{(t)}(\boldsymbol{x}) \right| \leq C \tilde{d}^{-1/4} \tag{10}$$

---

[4]$f$ is *Lipschitz* if there exists a constant $L > 0$ such that for any $\boldsymbol{x}_1, \boldsymbol{x}_2$, $|f(\boldsymbol{x}_1) - f(\boldsymbol{x}_2)| \leq L \|\boldsymbol{x}_1 - \boldsymbol{x}_2\|_2$.

[5]*Non-degenerated* means that $\Theta(\boldsymbol{x}, \boldsymbol{x}')$ depends on $\boldsymbol{x}$ and $\boldsymbol{x}'$ and is not a constant.

**Remark.** *We can easily extend this theorem to the case where there exists $\alpha_l > 0$ for each of $l = 2, \cdots, L$ such that $\tilde{d}_l / d_1 \to \alpha_l$ and $d_1 \to \infty$.*

Combining all the above results altogether, we achieve our main theorem:

**Theorem 6.** *Under the conditions of Theorem 5, there exists an $\eta_1 > 0$ such that if $\eta \leq \eta_1$ and $\nabla_\theta f^{(0)}(\boldsymbol{x}_1), \cdots, \nabla_\theta f^{(0)}(\boldsymbol{x}_n)$ are linearly independent, then as $\tilde{d} \to \infty$, for any test point $\boldsymbol{x} \in \mathbb{R}^d$ such that $\|\boldsymbol{x}\|_2 \leq 1$, with probability close to 1 over random initialization,*

$$\limsup_{t \to \infty} \left| f^{(t)}(\boldsymbol{x}) - f^{(t)}_{\mathrm{ERM}}(\boldsymbol{x}) \right| = O(\tilde{d}^{-1/4}) \to 0 \tag{11}$$

*where $f^{(t)}$ is trained by the reweighting algorithm and $f^{(t)}_{\mathrm{ERM}}$ is trained by ERM.*

The main theorem shows that at any test point $\boldsymbol{x}$, the gap between the function values of the two models converges to an infinitely small term, so the worst-group test performance of the reweighting algorithm will converge to the same level as ERM. Therefore, we have proved that for sufficiently wide fully-connected neural networks, reweighting algorithms always overfit.

Our key intuition tells us that the change in the model parameters always lies in an $n$-dimensional subspace. Thus, one possible way to improve the worst-group test performance is to enlarge this subspace by adding more training samples, e.g. via data augmentation or semi-supervised learning. However, even if we have more training samples, as long as the model is still overparameterized, and all $\nabla_\theta f^{(0)}(\boldsymbol{x}_i)$ are linearly independent, then our result still says that no reweighting algorithm can do better than ERM in the long run (though the performance of ERM itself might be improved).

Moreover, our theoretical results can explain the surprising empirical observation in Sagawa et al. (2020b) that removing some samples from the majority groups to match the group sizes can sometimes achieve even higher worst-group test performance than reweighting even though it wastes lots of data (see their Section 6). When training samples are removed, the model will converge to an interpolator of the smaller training set which is different from the interpolator of the original training set, so there is a chance that the performance of the new interpolator is actually higher.

## 4 DOES REGULARIZATION REALLY HELP?

In the previous section, we proved the pessimistic result that reweighting algorithms always overfit, i.e. in the long run their worst-group test performances always drop to the same level as ERM. And even if we use strong data augmentation or semi-supervised learning, reweighting algorithms still cannot outperform ERM if the training set is not sufficiently enlarged.

Sagawa et al. (2020a) proposed to tackle the overfitting problem of reweighting algorithms via regularization. In particular, they empirically demonstrated with experiments that large regularization is required to prevent reweighting algorithms such as group DRO from overfitting. With a large regularization, the model can maintain a high test worst-group performance, but it cannot obtain perfect training accuracy, in contrast to the case where no regularization is applied.

In this section, we study the necessary conditions for regularization to maintain high worst-group test performance. Specifically, we will show that regularization will not work if it is not large enough to prevent the model from obtaining nearly zero training error. In other words, lowering the training performance is the key to keeping a high worst-group test performance. Note that *the results in this section do not require Assumption 1*, so the results hold for all reweighting algorithms.

### 4.1 THEORETICAL ANALYSIS

Consider a reweighting algorithm with sample weights $q_i^{(t)}$. Following Sagawa et al. (2020a), we consider adding $L_2$ penalty to the weighted empirical risk (2):

$$\hat{\mathcal{R}}^\mu_{\boldsymbol{q}^{(t)}}(f) = \sum_{i=1}^n q_i^{(t)} \ell(f(\boldsymbol{x}_i), y_i) + \frac{\mu}{2} \left\| \theta - \theta^{(0)} \right\|_2^2 \tag{12}$$

Given that sufficiently wide neural networks can be approximated by linearized ones, we first focus on linearized neural networks. We will use the subscript "reg" to refer to a regularized model (which

is trained trained by minimizing the regularized risk (12)). Let $f_{\text{linreg}}^{(t)}$ be a regularized linearized neural network trained by some reweighting algorithm, and $f_{\text{linERM}}^{(t)}$ be an unregularized linearized neural network trained by ERM. As before, we consider training the models with gradient descent under the squared loss $\ell(\hat{y}, y) = \frac{1}{2}(\hat{y} - y)^2$. The following result shows that these two models are very close if $f_{\text{linreg}}^{(t)}$ can achieve low training error:

**Theorem 7.** *If there is a constant $M_0 > 0$ such that $\left\| \nabla_\theta f^{(0)}(\boldsymbol{x}) \right\|_2 \leq M_0$ for all $\|\boldsymbol{x}\|_2 \leq 1$, $\nabla_\theta f^{(0)}(\boldsymbol{x}_1), \cdots, \nabla_\theta f^{(0)}(\boldsymbol{x}_n)$ are linearly independent, and the empirical training risk of $f_{\text{linreg}}^{(t)}$ satisfies*

$$\limsup_{t \to \infty} \hat{\mathcal{R}}(f_{\text{linreg}}^{(t)}) < \epsilon, \tag{13}$$

*for some $\epsilon > 0$, then for any test point $\boldsymbol{x}$ such that $\|\boldsymbol{x}\|_2 \leq 1$ we have*

$$\limsup_{t \to \infty} \left| f_{\text{linreg}}^{(t)}(\boldsymbol{x}) - f_{\text{linERM}}^{(t)}(\boldsymbol{x}) \right| = O(\sqrt{\epsilon}). \tag{14}$$

The proof of this theorem also follows the key intuition: we can show that even with the $L_2$ penalty added, $\theta^{(t)} - \theta^{(0)}$ is still limited in a low-dimensional subspace. And although we cannot prove that $\theta^{(t)}$ always converges to the ERM interpolator, we can prove that it can get very close to that interpolator if its training error is very low, so the resulting model is very close to the ERM model.

Then, we can extend this result to sufficiently wide fully-connected neural networks:

**Theorem 8.** *If $\lambda^{\min} > 0$ and $\mu > 0$, then let $\eta^* = (\mu + \lambda^{\min} + \lambda^{\max})^{-1}$. For a wide fully-connected neural network $f_{\text{reg}}^{(t)}$ defined by (8) and (9) and satisfying Assumption 2, and any reweighting algorithm, if $d_1 = d_2 = \cdots = d_L = \tilde{d}$, $\eta \leq \eta^*$, $\nabla_\theta f^{(0)}(\boldsymbol{x}_1), \cdots, \nabla_\theta f^{(0)}(\boldsymbol{x}_n)$ are linearly independent, and the empirical training risk of $f_{\text{reg}}^{(t)}$ satisfies*

$$\limsup_{t \to \infty} \hat{\mathcal{R}}(f_{\text{reg}}^{(t)}) < \epsilon \tag{15}$$

*for some $\epsilon > 0$, then as $\tilde{d} \to \infty$, with probability close to 1 over random initialization, for any test point $\boldsymbol{x}$ such that $\|\boldsymbol{x}\|_2 \leq 1$ we have*

$$\limsup_{t \to \infty} \left| f_{\text{reg}}^{(t)}(\boldsymbol{x}) - f_{\text{ERM}}^{(t)}(\boldsymbol{x}) \right| = O(\tilde{d}^{-1/4} + \sqrt{\epsilon}) \to O(\sqrt{\epsilon}) \tag{16}$$

The result shows that a regularized model trained by any reweighting algorithm will get very close to an unregularized ERM model at any test point $\boldsymbol{x}$ if the training error of the former is nearly zero. Thus, regularization only helps when it is large enough to keep the training error of the model away from zero by a margin.

Our results explain the empirical observation of Sagawa et al. (2020a) that by using large regularization, the model can maintain a high worst-group test performance, but it cannot achieve perfect training accuracy. If smaller regularization is applied and the model can achieve nearly perfect training accuracy, then its worst-group test performance will still significantly drop.

### 4.2 EMPIRICAL STUDY

In this section, we validate our theoretical results above with experiments on Waterbirds and CelebA. We run ERM, IW and group DRO under different levels of weight decay for 500 epochs on Waterbirds and 250 epochs on CelebA. Note that we do not strictly follow our $L_2$ penalty formulation (12), but we study the $L_2$ weight decay regularization which is most widely used in practice. We repeat each experiment five times with different random seeds and report the 95% confidence interval of the mean average training and worst-group test accuracies of the last 10 training epochs in Table 1. To compare with early stopping, we also report the mean accuracies of epochs 11-20 with no regularization in blue. Moreover, we plot the average training and worst-group test accuracy curves throughout training for IW and Group DRO with one of the random seeds in Figure 2.

On both datasets, early stopping achieve the best performances. Particularly, on Waterbirds, there is no clear sign that regularization could help prevent overfitting. When the regularization is small,

Table 1: Mean average training accuracy and worst-group test accuracy (%) of the last 10 training epochs of ERM, IW and Group DRO under different levels of weight decay (WD). Each entry is Average training accuracy / Worst-group test accuracy. Blue entries are mean accuracies of epochs 11-20 with no weight decay. Each experiment is repeated five times with different random seeds.

| Dataset | WD | ERM | IW | Group DRO |
|---|---|---|---|---|
| Waterbirds | 0 | $100.0 \pm 0.0/56.3 \pm 1.8$ | $100.0 \pm 0.0/67.6 \pm 1.1$ | $100.0 \pm 0.0/64.5 \pm 1.6$ |
| | (11-20) | (Early stopping) | $92.4 \pm 0.4/83.7 \pm 0.6$ | $92.9 \pm 0.4/79.9 \pm 2.1$ |
| | 0.05 | | $100.0 \pm 0.0/71.0 \pm 1.9$ | $100.0 \pm 0.0/63.5 \pm 2.6$ |
| | 0.1 | | $100.0 \pm 0.0/67.7 \pm 0.7$ | $100.0 \pm 0.0/54.7 \pm 2.7$ |
| | 0.15 | | $99.0 \pm 0.7/53.7 \pm 2.7$ | $99.4 \pm 0.6/52.5 \pm 2.5$ |
| | 0.2 | | $91.6 \pm 2.0/35.9 \pm 6.9$ | $94.8 \pm 0.9/38.0 \pm 7.5$ |
| CelebA | 0 | $99.0 \pm 0.2/40.2 \pm 5.6$ | $99.4 \pm 0.1/42.7 \pm 1.7$ | $99.4 \pm 0.1/49.5 \pm 1.9$ |
| | (11-20) | (Early stopping) | $92.1 \pm 0.3/78.2 \pm 3.2$ | $90.5 \pm 0.5/85.2 \pm 1.7$ |
| | 0.01 | | $97.9 \pm 0.2/50.0 \pm 2.8$ | $96.5 \pm 0.5/67.2 \pm 1.7$ |
| | 0.03 | | $95.0 \pm 0.2/62.8 \pm 2.4$ | $88.9 \pm 1.1/83.1 \pm 2.2$ |
| | 0.1 | | $89.4 \pm 2.0/76.0 \pm 2.4$ | $75.1 \pm 9.5/50.6 \pm 15.9$ |

(a) IW: Avg Train Acc.   (b) IW: WG Test Acc.   (c) GDRO: Avg Train Acc.   (d) GDRO: WG Test Acc.

Figure 2: Average training accuracy and worst-group (WG) test accuracy of IW and Group DRO (GDRO) under different $L_2$ weight decay levels on CelebA.

the training accuracy is still 100% and the algorithm continues to overfit. However, when the regularization is large enough to lower the training accuracy, the worst-group test accuracy drops more because the model cannot learn the samples well under such a large regularization. Thus, perhaps not surprisingly, a lower training performance is only a necessary condition but not sufficient.

On CelebA, regularization does help mitigate overfitting, but a useful regularization must be large enough to lower the training accuracy. We observe that Group DRO overfits more slowly than IW, as it still has over 70% worst-group test accuracy after 70 epochs. However, as Figure 2d clearly shows, its worst-group test accuracy will still drop to the ERM level at 200 epochs. We also notice that Group DRO requires a smaller regularization than IW: for IW we need the weight decay level to be as large as 0.1 to achieve a similar performance as early stopping, but for Group DRO it only needs to be 0.01, and using 0.1 is actually harmful.

Overall, we find that early stopping achieves a markedly better performance. On the other hand, using large regularization could result in training instability, as well as a loss in overall performance, and there may or may not be a small band for the regularization parameter where the worst-group test performance is better.

## 5   CONCLUSION

In this work, we theoretically studied why reweighting algorithms overfit in practice by analyzing their implicit biases. Specifically, we proved the pessimistic result that reweighting algorithms always overfit. Our proof was based on the key intuition that the change in model parameters always lies in a low-dimensional subspace, so that even with reweighting, the model still converges to the same unique interpolator. When regularization is applied, we proved that the regularization must be large enough to keep the model from achieving nearly zero training error in order to prevent overfitting. We empirically validated our theoretical results on real datasets, and our results can also explain the empirical observations in previous work. Our results are especially important for large-scale machine learning tasks, where early stopping is not always possible in order to achieve high performances. Practitioners shooting for high worst-group performances in those tasks must be very careful about to what extent overfitting affects reweighting algorithms.

REPRODUCIBILITY STATEMENT

To guarantee the reproducibility of all our empirical results, in all our experiments we use a fixed set of random seeds, and we run some of the experiments twice with the same random seed to make sure that the outputs are the same. See Appendix D.1 for experiment details. After this paper is deanonymized, we will provide a GitHub repository that contains all the codes, datasets, hyperparameters, random seeds, machine speculations and anaconda environment speculations that are sufficient to exactly reproduce our empirical results.

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

## A    OTHER REWEIGHTING ALGORITHMS

In this section, we will review some other previously proposed reweighting algorithms. First, we will look at DRO-based methods, where DRO stands for Distributionally Robust Optimization.

DRO is designed for tasks with *distributional shift*, where the training distribution and the test distribution are different, and there are some constraints on the distance between these two distributions (typically described by a divergence function $D$). Since the real test distribution is unknown, DRO minimizes the model's risk over the worst distribution that satisfies the distance constraints, which is an upper bound of the model's real test error. Formally speaking, given a training distribution $P$, DRO minimizes the expected risk over the worst-case distribution $Q$ in a ball w.r.t. divergence $D$ around the training distribution $P$. For *group shift* problems which require high worst-group performance, $Q$ also needs to be absolutely continuous with respect to $P$, i.e. $Q \ll P$. Overall, DRO minimizes the following *expected DRO risk*:

$$\mathcal{R}_{D,\rho}(\theta; P) = \sup_{Q \ll P} \{\mathbb{E}_Q[\ell(\theta; Z)] : D(Q \parallel P) \leq \rho\} \tag{17}$$

The expected DRO risk is typically minimized in the following way: for each epoch $t$, we first find the worst $Q$ that maximizes $\mathbb{E}_Q[\ell(\theta; Z)]$ and satisfies $D(Q \parallel P) \leq \rho, Q \ll P$, and then minimize the model's expected risk over this $Q$ with gradient descent. The rationale behind this algorithm is the famous Danskin's Theorem, which says that if $F(x)$ is the maximum of a family of functions, then its gradient at point $x$ is equal to the gradient of the function that attains the maximum value at $x$.

Note that in practice we only have a finite set of training samples $\{z_1, \cdots, z_n\}$, so $P$ is always chosen as the empirical distribution, i.e. uniform distribution over $z_1, \cdots, z_n$. Then, note that $Q \ll P$, which implies that the support of $Q$ must be a subset of the support of $P$, which is $\{z_1, \cdots, z_n\}$. This means that $Q$ must be a distribution over $z_1, \cdots, z_n$, i.e. it is a reweighting over the training samples. Thus, we have essentially showed that DRO is a reweighting algorithm, and in fact almost all methods based on DRO are reweighting algorithms.

Two widely used variants of DRO are CVaR (Conditional Value at Risk) and $\chi^2$-DRO. In CVaR, for a fixed $\alpha \in (0, 1)$, we let $D(Q \parallel P) = \sup \log \frac{dQ}{dP}$ and $\rho = -\log \alpha$. As a result, suppose that $\alpha n$ is an integer, then CVaR will assign weight $\frac{1}{\alpha n}$ to $\alpha n$ training samples that incur the highest losses, and weight 0 to the rest of the samples, so we can easily see that CVaR is a reweighting algorithm. $\chi^2$-DRO was first used in Hashimoto et al. (2018) to deal with fairness tasks where the group labels are unknown, where $D(Q \parallel P) = \frac{1}{2} \int (dQ/dP - 1)^2 dP$ and $\rho = \frac{1}{2}(\frac{1}{\alpha} - 1)^2$. $\chi^2$-DRO is also a reweighting algorithm.

There are many other previously proposed methods of maximizing the worst-group performance that are also reweighting algorithms. For instance, Xu et al. (2020) studied the imbalanced class problem where a standard trained model always has high performance over classes with many training samples and low performance over minority classes. They proposed to balance the classes with Label CVaR, which is based on DRO and is a reweighting algorithm.

Liu et al. (2021) proposed a two-stage training process called JTT: in the first identification stage they trained a model with ERM to identify training samples that are hard to learn, and in the second upweighting stage they trained a new model with the hard samples upweighted, so that the model could learn all samples equally well. As the process itself suggests, JTT is a reweighting algorithm.

Finally, Zhai et al. (2021) argued that DRO-based methods are very sensitive to outliers in the training set because they upweight training samples with high losses and outliers tend to incur high losses. They proposed the DORO algorithm which at each iteration removes the samples with the highest losses, and then performs DRO on the rest of the samples. DORO is a reweighting algorithm.

## B    PROOFS

**Notations.**    In all of the proofs, for a matrix $\boldsymbol{A}$, we will use $\|\boldsymbol{A}\|_2$ to denote its spectral norm and $\|\boldsymbol{A}\|_F$ to denote its Frobenius norm.

## B.1 Proof of Theorem 1

To help our readers understand the proof more easily, we will first prove the result for static reweighting algorithms where $q_i^{(t)} = q_i$ for all $t$, and then we will prove the result for dynamic reweighting algorithms that satisfy $q_i^{(t)} \to q_i$ as $t \to \infty$.

### B.1.1 Static reweighting algorithms

We first prove the result for all static reweighting algorithms such that $\min_i q_i = q^* > 0$.

We will use a standard optimization proof technique called *smoothness*. Denote $A = \sum_{i=1}^n \|x_i\|_2^2$. The empirical risk of the linear model $f(x) = \langle \theta, x \rangle$ is

$$F(\theta) = \sum_{i=1}^n q_i (x_i^\top \theta - y_i)^2 \tag{18}$$

whose Hessian is

$$\nabla_\theta^2 F(\theta) = 2 \sum_{i=1}^n q_i x_i x_i^\top \tag{19}$$

So for any unit vector $v \in \mathbb{R}^d$, we have (since $q_i \in [0,1]$)

$$v^\top \nabla_\theta^2 F(\theta) v = 2 \sum_{i=1}^n q_i (x_i^\top v)^2 \leq 2 \sum_{i=1}^n q_i \|x_i\|_2^2 \leq 2A \tag{20}$$

which implies that $F(\theta)$ is $2A$-smooth. Thus, we have the following upper quadratic bound: for any $\theta_1, \theta_2 \in \mathbb{R}^d$,

$$F(\theta_2) \leq F(\theta_1) + \langle \nabla_\theta F(\theta_1), \theta_2 - \theta_1 \rangle + A \|\theta_2 - \theta_1\|_2^2 \tag{21}$$

Denote $g(\theta^{(t)}) = \sqrt{Q}(X^\top \theta^{(t)} - Y) \in \mathbb{R}^n$ where $\sqrt{Q} = \text{diag}(\sqrt{q_1}, \cdots, \sqrt{q_n})$. We can see that $\left\| g(\theta^{(t)}) \right\|_2^2 = F(\theta^{(t)})$, so that $\nabla F(\theta^{(t)}) = 2X\sqrt{Q}g(\theta^{(t)})$. The update rule of a static reweighting algorithm with gradient descent and the squared loss is:

$$\theta^{(t+1)} = \theta^{(t)} - \eta \sum_{i=1}^n q_i x_i (f^{(t)}(x_i) - y_i) = \theta^{(t)} - \eta X\sqrt{Q}g(\theta^{(t)}) \tag{22}$$

Substituting $\theta_1$ and $\theta_2$ in (21) with $\theta^{(t)}$ and $\theta^{(t+1)}$ yields

$$F(\theta^{(t+1)}) \leq F(\theta^{(t)}) - 2\eta g(\theta^{(t)})^\top \sqrt{Q}^\top X^\top X \sqrt{Q} g(\theta^{(t)}) + A \left\| \eta X \sqrt{Q} g(\theta^{(t)}) \right\|_2^2 \tag{23}$$

Since $x_1, \cdots, x_n$ are linearly independent, $X^\top X$ is a positive definite matrix. Denote the smallest eigenvalue of $X^\top X$ by $\lambda^{\min} > 0$. And $\left\| \sqrt{Q} g(\theta^{(t)}) \right\|_2 \geq \sqrt{q^*} \left\| g(\theta^{(t)}) \right\|_2 = \sqrt{q^* F(\theta^{(t)})}$, so we have $g(\theta^{(t)})^\top \sqrt{Q}^\top X^\top X \sqrt{Q} g(\theta^{(t)}) \geq q^* \lambda^{\min} F(\theta^{(t)})$. Thus,

$$\begin{aligned}
F(\theta^{(t+1)}) &\leq F(\theta^{(t)}) - 2\eta q^* \lambda^{\min} F(\theta^{(t)}) + A\eta^2 \left\| X\sqrt{Q} \right\|_2^2 \left\| g(\theta^{(t)}) \right\|_2^2 \\
&\leq F(\theta^{(t)}) - 2\eta q^* \lambda^{\min} F(\theta^{(t)}) + A\eta^2 \left\| X\sqrt{Q} \right\|_F^2 F(\theta^{(t)}) \\
&\leq F(\theta^{(t)}) - 2\eta q^* \lambda^{\min} F(\theta^{(t)}) + A\eta^2 \|X\|_F^2 F(\theta^{(t)}) \\
&= (1 - 2\eta q^* \lambda^{\min} + A^2 \eta^2) F(\theta^{(t)})
\end{aligned} \tag{24}$$

Let $\eta_0 = \frac{q^* \lambda^{\min}}{A^2}$. For any $\eta \leq \eta_0$, we have $F(\theta^{(t+1)}) \leq (1 - \eta q^* \lambda^{\min}) F(\theta^{(t)})$ for all $t$, which implies that $\lim_{t \to \infty} F(\theta^{(t)}) = 0$. Moreover, $\sqrt{F(\theta^{(t+1)})} \leq (1 - \frac{\eta q^* \lambda^{\min}}{2}) \sqrt{F(\theta^{(t)})}$ due to $\sqrt{1-x} \leq 1 - x/2$.

The convergence in $F(\theta)$ implies the convergence in $\theta$. This is because

$$\left\|\theta^{(t+1)} - \theta^{(t)}\right\|_2^2 = \eta^2 \left\|\boldsymbol{X}\sqrt{\boldsymbol{Q}}g(\theta^{(t)})\right\|_2^2 \le \eta^2 \left\|\boldsymbol{X}\sqrt{\boldsymbol{Q}}\right\|_F^2 \left\|g(\theta^{(t)})\right\|_2^2$$
$$\le \eta^2 \left\|\boldsymbol{X}\right\|_F^2 \left\|g(\theta^{(t)})\right\|_2^2 = A\eta^2 F(\theta^{(t)}) \tag{25}$$

which implies that for any $\eta \le \eta_0$,

$$\sum_{t=T}^{\infty} \left\|\theta^{(t+1)} - \theta^{(t)}\right\|_2 \le \sqrt{A\eta^2} \sum_{t=T}^{\infty} \sqrt{F(\theta^{(t)})} \le \frac{2A}{q^*\lambda^{\min}}\sqrt{F(\theta^{(T)})} \tag{26}$$

Therefore, $\lim_{T\to\infty}\sum_{t=T}^{\infty}\left\|\theta^{(t+1)} - \theta^{(t)}\right\|_2 = 0$, which means that $\theta^{(t)}$ converges, and it converges to some interpolator.

### B.1.2 DYNAMIC REWEIGHTING ALGORITHMS

Now we prove the result for all dynamic reweighting algorithms satisfying Assumption1. By Assumption 1, for any $\epsilon > 0$, there exists $t_\epsilon$ such that for all $t \ge t_\epsilon$ and all $i$,

$$q_i^{(t)} \in (q_i - \epsilon, q_i + \epsilon) \tag{27}$$

This is because for all $i$, there exists $t_i$ such that for all $t \ge t_i$, $q_i^{(t)} \in (q_i - \epsilon, q_i + \epsilon)$. Then, we can define $t_\epsilon = \max\{t_1, \cdots, t_n\}$. Denote the largest and smallest eigenvalues of $\boldsymbol{X}^\top\boldsymbol{X}$ by $\lambda^{\max}$ and $\lambda^{\min}$, and because $\boldsymbol{X}$ is full-rank, we have $\lambda^{\min} > 0$. Select and a fix an $\epsilon$ such that $0 < \epsilon < \max\{\frac{q^*}{3}, \frac{(q^*\lambda^{\min})^2}{12\lambda^{\max 2}}\}$, and then $t_\epsilon$ is also fixed.

We still denote $\boldsymbol{Q} = \operatorname{diag}(q_1, \cdots, q_n)$. When $t \ge t_\epsilon$, the update rule of a dynamic reweighting algorithm with gradient descent and the squared loss is:

$$\theta^{(t+1)} = \theta^{(t)} - \eta\boldsymbol{X}\boldsymbol{Q}_\epsilon^{(t)}(\boldsymbol{X}^\top\theta^{(t)} - \boldsymbol{Y}) \tag{28}$$

where $\boldsymbol{Q}_\epsilon^{(t)} = \boldsymbol{Q}^{(t)}$, and we use the subscript $\epsilon$ to indicate that $\left\|\boldsymbol{Q}_\epsilon^{(t)} - \boldsymbol{Q}\right\|_2 < \epsilon$. Then, note that we can rewrite $\boldsymbol{Q}_\epsilon^{(t)}$ as $\boldsymbol{Q}_\epsilon^{(t)} = \sqrt{\boldsymbol{Q}_{3\epsilon}^{(t)}}\cdot\sqrt{\boldsymbol{Q}}$ for all $\epsilon < q^*/3$. This is because $q_i + \epsilon < \sqrt{(q_i + 3\epsilon)q_i}$ and $q_i - \epsilon > \sqrt{(q_i - 3\epsilon)q_i}$ for all $\epsilon < q_i/3$, and $q_i \ge q^*$. Thus, we have

$$\theta^{(t+1)} = \theta^{(t)} - \eta\boldsymbol{X}\sqrt{\boldsymbol{Q}_{3\epsilon}^{(t)}}g(\theta^{(t)}) \quad \text{where } \boldsymbol{Q}_\epsilon^{(t)} = \sqrt{\boldsymbol{Q}_{3\epsilon}^{(t)}}\cdot\sqrt{\boldsymbol{Q}} \tag{29}$$

Again, substituting $\theta_1$ and $\theta_2$ in (21) with $\theta^{(t)}$ and $\theta^{(t+1)}$ yields

$$F(\theta^{(t+1)}) \le F(\theta^{(t)}) - 2\eta g(\theta^{(t)})^\top\sqrt{\boldsymbol{Q}}^\top\boldsymbol{X}^\top\boldsymbol{X}\sqrt{\boldsymbol{Q}_{3\epsilon}^{(t)}}g(\theta^{(t)}) + A\left\|\eta\boldsymbol{X}\sqrt{\boldsymbol{Q}_{3\epsilon}^{(t)}}g(\theta^{(t)})\right\|_2^2 \tag{30}$$

Then, note that

$$\left|g(\theta^{(t)})^\top\sqrt{\boldsymbol{Q}}^\top\boldsymbol{X}^\top\boldsymbol{X}\left(\sqrt{\boldsymbol{Q}_{3\epsilon}^{(t)}} - \sqrt{\boldsymbol{Q}}\right)g(\theta^{(t)})\right|$$
$$\le \left\|\sqrt{\boldsymbol{Q}}^\top\boldsymbol{X}^\top\boldsymbol{X}\left(\sqrt{\boldsymbol{Q}_{3\epsilon}^{(t)}} - \sqrt{\boldsymbol{Q}}\right)\right\|_2 \left\|g(\theta^{(t)})\right\|_2^2$$
$$\le \left\|\sqrt{\boldsymbol{Q}}\right\|_2 \left\|\boldsymbol{X}^\top\boldsymbol{X}\right\|_2 \left\|\sqrt{\boldsymbol{Q}_{3\epsilon}^{(t)}} - \sqrt{\boldsymbol{Q}}\right\|_2 \left\|g(\theta^{(t)})\right\|_2^2 \tag{31}$$
$$\le \lambda^{\max}\sqrt{3\epsilon}F(\theta^{(t)})$$

where the last step comes from the following fact: for all $\epsilon < q_i/3$,

$$\sqrt{q_i + 3\epsilon} - \sqrt{q_i} \le \sqrt{3\epsilon} \quad \text{and} \quad \sqrt{q_i} - \sqrt{q_i - 3\epsilon} \le \sqrt{3\epsilon} \tag{32}$$

And as proved before, we also have

$$g(\theta^{(t)})^\top \sqrt{Q}^\top X^\top X \sqrt{Q} g(\theta^{(t)}) \geq q^* \lambda^{\min} F(\theta^{(t)}) \tag{33}$$

Since $\epsilon \leq \frac{(q^* \lambda^{\min})^2}{12\lambda^{\max 2}}$, we have

$$g(\theta^{(t)})^\top \sqrt{Q}^\top X^\top X \sqrt{Q_{3\epsilon}^{(t)}} g(\theta^{(t)}) \geq \left( q^* \lambda^{\min} - \lambda^{\max} \sqrt{3\epsilon} \right) F(\theta^{(t)}) \geq \frac{1}{2} q^* \lambda^{\min} F(\theta^{(t)}) \tag{34}$$

Thus,

$$\begin{aligned} F(\theta^{(t+1)}) &\leq F(\theta^{(t)}) - \eta q^* \lambda^{\min} F(\theta^{(t)}) + A\eta^2 \left\| X \sqrt{Q_{3\epsilon}^{(t)}} \right\|_2^2 \left\| g(\theta^{(t)}) \right\|_2^2 \\ &\leq (1 - \eta q^* \lambda^{\min} + A^2 \eta^2 (1 + 3\epsilon)) F(\theta^{(t)}) \\ &\leq (1 - \eta q^* \lambda^{\min} + 2A^2 \eta^2) F(\theta^{(t)}) \end{aligned} \tag{35}$$

for all $\epsilon < 1/3$. Let $\eta_0 = \frac{q^* \lambda^{\min}}{4A^2}$. For any $\eta \leq \eta_0$, we have $F(\theta^{(t+1)}) \leq (1 - \eta q^* \lambda^{\min}/2) F(\theta^{(t)})$ for all $t \geq t_\epsilon$, which implies that $\lim_{t\to\infty} F(\theta^{(t)}) = 0$. As before, we can prove that the convergence in $F(\theta)$ implies the convergence in $\theta$. Thus, $\theta$ converges to some interpolator. $\square$

## B.2 PROOF OF THEOREM 4

Note that the first $l$ layers (except the output layer) of the original NTK formulation and our new formulation are the same, so we still have the following proposition:

**Proposition 9** (Proposition 1 in Jacot et al. (2018)). *If $\sigma$ is Lipschitz and $d_l \to \infty$ for $l = 1, \cdots, L$ sequentially, then for all $l = 1, \cdots, L$, the distribution of a single element of $h^l$ converges in probability to a zero-mean Gaussian process of covariance $\Sigma^l$ that is defined recursively by:*

$$\begin{aligned} \Sigma^1(x, x') &= \frac{1}{d_0} x^\top x' + \beta^2 \\ \Sigma^l(x, x') &= \mathbb{E}_f [\sigma(f(x))\sigma(f(x'))] + \beta^2 \end{aligned} \tag{36}$$

*where $f$ is sampled from a zero-mean Gaussian process of covariance $\Sigma^{(l-1)}$.*

Now we show that for an infinitely wide neural network with $L \geq 1$ hidden layers, $\Theta^{(0)}$ converges in probability to the following non-degenerated deterministic limiting kernel

$$\Theta = \mathbb{E}_{f \sim \Sigma^L} [\sigma(f(x))\sigma(f(x'))] + \beta^2 \tag{37}$$

Consider the output layer $h^{L+1} = \frac{W^L}{\sqrt{\tilde{d}}} \sigma(h^L) + \beta b^L$. We can see that for any parameter $\theta_i$ before the output layer,

$$\nabla_{\theta_i} h^{L+1} = \operatorname{diag}(\dot{\sigma}(h^L)) \frac{W^{L\top}}{\sqrt{d_L}} \nabla_{\theta_i} h^L = 0 \tag{38}$$

And for $W^L$ and $b^L$, we have

$$\nabla_{W^L} h^{L+1} = \frac{1}{\sqrt{d_L}} \sigma(h^L) \qquad \text{and} \qquad \nabla_{b^L} h^{L+1} = \beta \tag{39}$$

Then we can achieve (37) by the law of large numbers. $\square$

## B.3 PROOF OF THEOREM 5

We will use the following short-hand in the proof:

$$\begin{cases} g(\theta^{(t)}) = f^{(t)}(X) - Y \\ J(\theta^{(t)}) = \nabla_\theta f(X; \theta^{(t)}) \in \mathbb{R}^{p \times n} \\ \Theta^{(t)} = J(\theta^{(t)})^\top J(\theta^{(t)}) \end{cases} \tag{40}$$

For any $\epsilon > 0$, there exists $t_\epsilon$ such that for all $t \geq t_\epsilon$ and all $i$, $q_i^{(t)} \in (q_i - \epsilon, q_i + \epsilon)$. Like what we have done in (29), we can rewrite $\boldsymbol{Q}^{(t)} = \boldsymbol{Q}_\epsilon^{(t)} = \sqrt{\boldsymbol{Q}_{3\epsilon}^{(t)}} \cdot \sqrt{\boldsymbol{Q}}$, where $\boldsymbol{Q} = \text{diag}(q_1, \cdots, q_n)$.

The update rule of a reweighting algorithm with gradient descent and the squared loss for the wide neural network is:

$$\theta^{(t+1)} = \theta^{(t)} - \eta J(\theta^{(t)}) \boldsymbol{Q}^{(t)} g(\theta^{(t)}) \tag{41}$$

and for $t \geq t_\epsilon$, it can be rewritten as

$$\theta^{(t+1)} = \theta^{(t)} - \eta J(\theta^{(t)}) \sqrt{\boldsymbol{Q}_{3\epsilon}^{(t)}} \left[ \sqrt{\boldsymbol{Q}} g(\theta^{(t)}) \right] \tag{42}$$

First, we will prove the following theorem:

**Theorem 10.** *There exist constants $M > 0$ and $\epsilon_0 > 0$ such that for all $\epsilon \in (0, \epsilon_0]$, $\eta \leq \eta^*$ and any $\delta > 0$, there exist $R_0 > 0$, $\tilde{D} > 0$ and $B > 1$ such that for any $\tilde{d} \geq \tilde{D}$, the following (i) and (ii) hold with probability at least $(1 - \delta)$ over random initialization when applying gradient descent with learning rate $\eta$:*

*(i) For all $t \leq t_\epsilon$, there is*

$$\left\| g(\theta^{(t)}) \right\|_2 \leq B^t R_0 \tag{43}$$

$$\sum_{j=1}^t \left\| \theta^{(j)} - \theta^{(j-1)} \right\|_2 \leq \eta M R_0 \sum_{j=1}^t B^{j-1} < \frac{MB^{t_\epsilon} R_0}{B - 1} \tag{44}$$

*(ii) For all $t \geq t_\epsilon$, we have*

$$\left\| \sqrt{\boldsymbol{Q}} g(\theta^{(t)}) \right\|_2 \leq \left( 1 - \frac{\eta q^* \lambda^{\min}}{3} \right)^{t - t_\epsilon} B^{t_\epsilon} R_0 \tag{45}$$

$$\sum_{j=t_\epsilon+1}^t \left\| \theta^{(j)} - \theta^{(j-1)} \right\|_2 \leq \eta \sqrt{1 + 3\epsilon} M B^{t_\epsilon} R_0 \sum_{j=t_\epsilon+1}^t \left( 1 - \frac{\eta q^* \lambda^{\min}}{3} \right)^{j - t_\epsilon}$$

$$< \frac{3\sqrt{1 + 3\epsilon} M B^{t_\epsilon} R_0}{q^* \lambda^{\min}} \tag{46}$$

*Proof.* The proof is based on the following lemma:

**Lemma 11** (Local Lipschitzness of the Jacobian). *Under Assumption 2, there is a constant $M > 0$ such that for any $C_0 > 0$ and any $\delta > 0$, there exists a $\tilde{D}$ such that: If $\tilde{d} \geq \tilde{D}$, then with probability at least $(1 - \delta)$ over random initialization, for any $\boldsymbol{x}$ such that $\|\boldsymbol{x}\|_2 \leq 1$,*

$$\begin{cases} \left\| \nabla_\theta f(\boldsymbol{x}; \theta) - \nabla_\theta f(\boldsymbol{x}; \tilde{\theta}) \right\|_2 \leq \frac{M}{\sqrt[4]{\tilde{d}}} \left\| \theta - \tilde{\theta} \right\|_2 \\ \left\| \nabla_\theta f(\boldsymbol{x}; \theta) \right\|_2 \leq M \\ \left\| J(\theta) - J(\tilde{\theta}) \right\|_F \leq \frac{M}{\sqrt[4]{\tilde{d}}} \left\| \theta - \tilde{\theta} \right\|_2 \\ \left\| J(\theta) \right\|_F \leq M \end{cases} \quad \forall \theta, \tilde{\theta} \in B(\theta^{(0)}, C_0) \tag{47}$$

*where $B(\theta^{(0)}, R) = \{\theta : \left\| \theta - \theta^{(0)} \right\|_2 < R\}$.*

The proof can be found in Appendix B.4. Note that for any $\boldsymbol{x}$, $f^{(0)}(\boldsymbol{x}) = \beta \boldsymbol{b}^L$ where $\boldsymbol{b}^L$ is sampled from the standard Gaussian distribution. Thus, for any $\delta > 0$, there exists a constant $R_0$ such that with probability at least $(1 - \delta/3)$ over random initialization,

$$\left\| g(\theta^{(0)}) \right\|_2 < R_0 \tag{48}$$

And by Theorem 4, there exists $D_2 \geq 0$ such that for any $\tilde{d} \geq D_2$, with probability at least $(1 - \delta/3)$,

$$\left\| \Theta - \Theta^{(0)} \right\|_F \leq \frac{q^* \lambda^{\min}}{3} \tag{49}$$

Let $M$ be the constant in Lemma 11. Let $\epsilon_0 = \frac{(q^* \lambda^{\min})^2}{108 M^4}$. Let $B = 1 + \eta^* M^2$, and $C_0 = \frac{M B^{t_\epsilon} R_0}{B-1} + \frac{3\sqrt{1+3\epsilon} M B^{t_\epsilon} R_0}{q^* \lambda^{\min}}$. By Lemma 11, there exists $D_1 > 0$ such that with probability at least $(1 - \delta/3)$, for any $\tilde{d} \geq D_1$, (47) is true for all $\theta, \tilde{\theta} \in B(\theta^{(0)}, C_0)$.

By union bound, with probability at least $(1 - \delta)$, (47), (48) and (49) are all true. Now we assume that all of them are true, and prove (43) and (44) by induction. (43) is true for $t = 0$ due to (48), and (44) is always true for $t = 0$. Suppose (43) and (44) are true for $t$, then for $t + 1$ we have

$$
\begin{aligned}
\left\| \theta^{(t+1)} - \theta^{(t)} \right\|_2 &\leq \eta \left\| J(\theta^{(t)}) \boldsymbol{Q}^{(t)} \right\|_2 \left\| g(\theta^{(t)}) \right\|_2 \leq \eta \left\| J(\theta^{(t)}) \boldsymbol{Q}^{(t)} \right\|_F \left\| g(\theta^{(t)}) \right\|_2 \\
&\leq \eta \left\| J(\theta^{(t)}) \right\|_F \left\| g(\theta^{(t)}) \right\|_2 \leq M \eta B^t R_0
\end{aligned}
\tag{50}
$$

So (44) is also true for $t + 1$. And we also have

$$
\begin{aligned}
\left\| g(\theta^{(t+1)}) \right\|_2 &= \left\| g(\theta^{(t+1)}) - g(\theta^{(t)}) + g(\theta^{(t)}) \right\|_2 \\
&= \left\| J(\tilde{\theta}^{(t)})^\top (\theta^{(t+1)} - \theta^{(t)}) + g(\theta^{(t)}) \right\|_2 \\
&= \left\| -\eta J(\tilde{\theta}^{(t)})^\top J(\theta^{(t)}) \boldsymbol{Q}^{(t)} g(\theta^{(t)}) + g(\theta^{(t)}) \right\|_2 \\
&\leq \left\| \boldsymbol{I} - \eta J(\tilde{\theta}^{(t)})^\top J(\theta^{(t)}) \boldsymbol{Q}^{(t)} \right\|_2 \left\| g(\theta^{(t)}) \right\|_2 \\
&\leq \left( 1 + \left\| \eta J(\tilde{\theta}^{(t)})^\top J(\theta^{(t)}) \boldsymbol{Q}^{(t)} \right\|_2 \right) \left\| g(\theta^{(t)}) \right\|_2 \\
&\leq \left( 1 + \eta \left\| J(\tilde{\theta}^{(t)}) \right\|_F \left\| J(\theta^{(t)}) \right\|_F \right) \left\| g(\theta^{(t)}) \right\|_2 \\
&\leq (1 + \eta^* M^2) \left\| g(\theta^{(t)}) \right\|_2 \leq B^{t+1} R_0
\end{aligned}
\tag{51}
$$

Therefore, (43) and (44) are true for all $t \leq t_\epsilon$, which implies that $\left\| \sqrt{\boldsymbol{Q}} g(\theta^{(t_\epsilon)}) \right\|_2 \leq \left\| g(\theta^{(t_\epsilon)}) \right\|_2 \leq B^{t_\epsilon} R_0$, so (45) is true for $t = t_\epsilon$. And (46) is obviously true for $t = t_\epsilon$. Now, let us prove (ii) by induction. Note that when $t \geq t_\epsilon$, we have the alternative update rule (42). If (45) and (46) are true for $t$, then for $t + 1$, there is

$$
\begin{aligned}
\left\| \theta^{(t+1)} - \theta^{(t)} \right\|_2 &\leq \eta \left\| J(\theta^{(t)}) \sqrt{\boldsymbol{Q}_{3\epsilon}^{(t)}} \right\|_2 \left\| \sqrt{\boldsymbol{Q}} g(\theta^{(t)}) \right\|_2 \leq \eta \left\| J(\theta^{(t)}) \sqrt{\boldsymbol{Q}_{3\epsilon}^{(t)}} \right\|_F \left\| \sqrt{\boldsymbol{Q}} g(\theta^{(t)}) \right\|_2 \\
&\leq \eta \sqrt{1 + 3\epsilon} \left\| J(\theta^{(t)}) \right\|_F \left\| \sqrt{\boldsymbol{Q}} g(\theta^{(t)}) \right\|_2 \leq M \eta \sqrt{1 + 3\epsilon} \left( 1 - \frac{\eta q^* \lambda^{\min}}{3} \right)^{t - t_\epsilon} B^{t_\epsilon} R_0
\end{aligned}
\tag{52}
$$

So (46) is true for $t + 1$. And we also have

$$
\begin{aligned}
\left\| \sqrt{\boldsymbol{Q}} g(\theta^{(t+1)}) \right\|_2 &= \left\| \sqrt{\boldsymbol{Q}} g(\theta^{(t+1)}) - \sqrt{\boldsymbol{Q}} g(\theta^{(t)}) + \sqrt{\boldsymbol{Q}} g(\theta^{(t)}) \right\|_2 \\
&= \left\| \sqrt{\boldsymbol{Q}} J(\tilde{\theta}^{(t)})^\top (\theta^{(t+1)} - \theta^{(t)}) + \sqrt{\boldsymbol{Q}} g(\theta^{(t)}) \right\|_2 \\
&= \left\| -\eta \sqrt{\boldsymbol{Q}} J(\tilde{\theta}^{(t)})^\top J(\theta^{(t)}) \boldsymbol{Q}^{(t)} g(\theta^{(t)}) + \sqrt{\boldsymbol{Q}} g(\theta^{(t)}) \right\|_2 \\
&\leq \left\| \boldsymbol{I} - \eta \sqrt{\boldsymbol{Q}} J(\tilde{\theta}^{(t)})^\top J(\theta^{(t)}) \sqrt{\boldsymbol{Q}_{3\epsilon}^{(t)}} \right\|_2 \left\| \sqrt{\boldsymbol{Q}} g(\theta^{(t)}) \right\|_2 \\
&\leq \left\| \boldsymbol{I} - \eta \sqrt{\boldsymbol{Q}} J(\tilde{\theta}^{(t)})^\top J(\theta^{(t)}) \sqrt{\boldsymbol{Q}_{3\epsilon}^{(t)}} \right\|_2 \left( 1 - \frac{\eta q^* \lambda^{\min}}{3} \right)^t R_0
\end{aligned}
\tag{53}
$$

where $\tilde{\theta}^{(t)}$ is some linear interpolation between $\theta^{(t)}$ and $\theta^{(t+1)}$. Now we prove that

$$
\left\| \boldsymbol{I} - \eta \sqrt{\boldsymbol{Q}} J(\tilde{\theta}^{(t)})^\top J(\theta^{(t)}) \sqrt{\boldsymbol{Q}_{3\epsilon}^{(t)}} \right\|_2 \leq 1 - \frac{\eta q^* \lambda^{\min}}{3}
\tag{54}
$$

For any unit vector $\boldsymbol{v} \in \mathbb{R}^n$, we have

$$\boldsymbol{v}^\top (\boldsymbol{I} - \eta \sqrt{\boldsymbol{Q}} \Theta \sqrt{\boldsymbol{Q}}) \boldsymbol{v} = 1 - \eta \boldsymbol{v}^\top \sqrt{\boldsymbol{Q}} \Theta \sqrt{\boldsymbol{Q}} \boldsymbol{v} \tag{55}$$

$\left\| \sqrt{\boldsymbol{Q}} \boldsymbol{v} \right\|_2 \in [\sqrt{q^*}, 1]$, so for any $\eta \leq \eta^*$, $\boldsymbol{v}^\top (\boldsymbol{I} - \eta \sqrt{\boldsymbol{Q}} \Theta \sqrt{\boldsymbol{Q}}) \boldsymbol{v} \in [0, 1 - \eta \lambda^{\min} q^*]$, which implies that $\left\| \boldsymbol{I} - \eta \sqrt{\boldsymbol{Q}} \Theta \sqrt{\boldsymbol{Q}} \right\|_2 \leq 1 - \eta \lambda^{\min} q^*$. Thus,

$$\left\| \boldsymbol{I} - \eta \sqrt{\boldsymbol{Q}} J(\tilde{\theta}^{(t)})^\top J(\theta^{(t)}) \sqrt{\boldsymbol{Q}} \right\|_2$$
$$\leq \left\| \boldsymbol{I} - \eta \sqrt{\boldsymbol{Q}} \Theta \sqrt{\boldsymbol{Q}} \right\|_2 + \eta \left\| \sqrt{\boldsymbol{Q}} (\Theta - \Theta^{(0)}) \sqrt{\boldsymbol{Q}} \right\|_2 + \eta \left\| \sqrt{\boldsymbol{Q}} (J(\theta^{(0)})^\top J(\theta^{(0)}) - J(\tilde{\theta}^{(t)})^\top J(\theta^{(t)})) \sqrt{\boldsymbol{Q}} \right\|_2$$
$$\leq 1 - \eta \lambda^{\min} q^* + \eta \left\| \sqrt{\boldsymbol{Q}} (\Theta - \Theta^{(0)}) \sqrt{\boldsymbol{Q}} \right\|_F + \eta \left\| \sqrt{\boldsymbol{Q}} (J(\theta^{(0)})^\top J(\theta^{(0)}) - J(\tilde{\theta}^{(t)})^\top J(\theta^{(t)})) \sqrt{\boldsymbol{Q}} \right\|_F$$
$$\leq 1 - \eta \lambda^{\min} q^* + \eta \left\| \Theta - \Theta^{(0)} \right\|_F + \eta \left\| J(\theta^{(0)})^\top J(\theta^{(0)}) - J(\tilde{\theta}^{(t)})^\top J(\theta^{(t)}) \right\|_F$$
$$\leq 1 - \eta \lambda^{\min} q^* + \frac{\eta q^* \lambda^{\min}}{3} + \frac{\eta M^2}{\sqrt[4]{\tilde{d}}} \left( \left\| \theta^{(t)} - \theta^{(0)} \right\|_2 + \left\| \tilde{\theta}^{(t)} - \theta^{(0)} \right\|_2 \right) \leq 1 - \frac{\eta q^* \lambda^{\min}}{2} \tag{56}$$

for all $\tilde{d} \geq \max \left\{ D_1, D_2, \left( \frac{12 M^2 C_0}{q^* \lambda^{\min}} \right)^4 \right\}$, which implies that

$$\left\| \boldsymbol{I} - \eta \sqrt{\boldsymbol{Q}} J(\tilde{\theta}^{(t)})^\top J(\theta^{(t)}) \sqrt{\boldsymbol{Q}_{3\epsilon}^{(t)}} \right\|_2$$
$$\leq 1 - \frac{\eta q^* \lambda^{\min}}{2} + \left\| \eta \sqrt{\boldsymbol{Q}} J(\tilde{\theta}^{(t)})^\top J(\theta^{(t)}) \left( \sqrt{\boldsymbol{Q}_{3\epsilon}^{(t)}} - \sqrt{\boldsymbol{Q}} \right) \right\|_2 \tag{57}$$
$$\leq 1 - \frac{\eta q^* \lambda^{\min}}{2} + \eta M^2 \sqrt{3\epsilon} \leq 1 - \frac{\eta q^* \lambda^{\min}}{3} \qquad \text{(due to (32))}$$

for all $\epsilon \leq \epsilon_0$. Thus, (45) is also true for $t + 1$. In conclusion, (45) and (46) are true with probability at least $(1 - \delta)$ for all $\tilde{d} \geq \tilde{D} = \max \left\{ D_1, D_2, \left( \frac{12 M^2 C_0}{q^* \lambda^{\min}} \right)^4 \right\}$. $\qquad \square$

Returning back to the proof of Theorem 5. Choose and fix an $\epsilon$ such that $\epsilon < \min\{\epsilon_0, \frac{1}{3} \left( \frac{q^* \lambda^{\min}}{3 \lambda^{\max} + q^* \lambda^{\min}} \right)^2\}$, where $\epsilon_0$ is defined by Theorem 10. Then, $t_\epsilon$ is also fixed. There exists $\tilde{D} \geq 0$ such that for any $\tilde{d} \geq \tilde{D}$, with probability at least $(1 - \delta)$, Theorem 10 and Lemma 11 are true and

$$\left\| \Theta - \Theta^{(0)} \right\|_F \leq \frac{q^* \lambda^{\min}}{3} \tag{58}$$

which immediately implies that

$$\left\| \Theta^{(0)} \right\|_2 \leq \left\| \Theta \right\|_2 + \left\| \Theta - \Theta^{(0)} \right\|_F \leq \lambda^{\max} + \frac{q^* \lambda^{\min}}{3} \tag{59}$$

We still denote $B = 1 + \eta^* M^2$ and $C_0 = \frac{M B^{t_\epsilon} R_0}{B - 1} + \frac{3\sqrt{1 + 3\epsilon} M B^{t_\epsilon} R_0}{q^* \lambda^{\min}}$. Theorem 10 ensures that for all $t$, $\theta^{(t)} \in B(\theta^{(0)}, C_0)$. Then we have

$$\left\| \boldsymbol{I} - \eta \sqrt{\boldsymbol{Q}} \Theta^{(0)} \sqrt{\boldsymbol{Q}} \right\|_2 \leq \left\| \boldsymbol{I} - \eta \sqrt{\boldsymbol{Q}} \Theta \sqrt{\boldsymbol{Q}} \right\|_2 + \eta \left\| \sqrt{\boldsymbol{Q}} (\Theta - \Theta^{(0)}) \sqrt{\boldsymbol{Q}} \right\|_2$$
$$\leq 1 - \eta \lambda^{\min} q^* + \frac{\eta q^* \lambda^{\min}}{3} = 1 - \frac{2\eta q^* \lambda^{\min}}{3} \tag{60}$$

so it follows that

$$\left\| \boldsymbol{I} - \eta \sqrt{\boldsymbol{Q}} \Theta^{(0)} \sqrt{\boldsymbol{Q}_{3\epsilon}^{(t)}} \right\|_2 \leq \left\| \boldsymbol{I} - \eta \sqrt{\boldsymbol{Q}} \Theta^{(0)} \sqrt{\boldsymbol{Q}} \right\|_2 + \left\| \eta \sqrt{\boldsymbol{Q}} \Theta^{(0)} \left( \sqrt{\boldsymbol{Q}_{3\epsilon}^{(t)}} - \sqrt{\boldsymbol{Q}} \right) \right\|_2 \tag{61}$$
$$\leq 1 - \frac{2\eta q^* \lambda^{\min}}{3} + \eta (\lambda^{\max} + \frac{q^* \lambda^{\min}}{3}) \sqrt{3\epsilon}$$

Thus, for all $\epsilon < \frac{1}{3}\left(\frac{q^*\lambda^{\min}}{3\lambda^{\max}+q^*\lambda^{\min}}\right)^2$, there is

$$\left\|\boldsymbol{I} - \eta\sqrt{\boldsymbol{Q}}\Theta^{(0)}\sqrt{\boldsymbol{Q}_{3\epsilon}^{(t)}}\right\|_2 \leq 1 - \frac{\eta q^*\lambda^{\min}}{3} \tag{62}$$

The update rule of the reweighting algorithm for the linearized neural network is:

$$\theta_{\mathrm{lin}}^{(t+1)} = \theta_{\mathrm{lin}}^{(t)} - \eta J(\theta^{(0)})\boldsymbol{Q}^{(t)}g_{\mathrm{lin}}(\theta^{(t)}) \tag{63}$$

where we use the subscript "lin" to denote the linearized neural network, and with a slight abuse of notion denote $g_{\mathrm{lin}}(\theta^{(t)}) = g(\theta_{\mathrm{lin}}^{(t)})$.

First, let us consider the training data $\boldsymbol{X}$. Denote $\Delta_t = g_{\mathrm{lin}}(\theta^{(t)}) - g(\theta^{(t)})$. We have

$$\begin{cases} g_{\mathrm{lin}}(\theta^{(t+1)}) - g_{\mathrm{lin}}(\theta^{(t)}) = -\eta J(\theta^{(0)})^\top J(\theta^{(0)})\boldsymbol{Q}^{(t)}g_{\mathrm{lin}}(\theta^{(t)}) \\ g(\theta^{(t+1)}) - g(\theta^{(t)}) = -\eta J(\tilde{\theta}^{(t)})^\top J(\theta^{(t)})\boldsymbol{Q}^{(t)}g(\theta^{(t)}) \end{cases} \tag{64}$$

where $\tilde{\theta}^{(t)}$ is some linear interpolation between $\theta^{(t)}$ and $\theta^{(t+1)}$. Thus,

$$\begin{aligned} \Delta_{t+1} - \Delta_t =& \eta\left[J(\tilde{\theta}^{(t)})^\top J(\theta^{(t)}) - J(\theta^{(0)})^\top J(\theta^{(0)})\right]\boldsymbol{Q}^{(t)}g(\theta^{(t)}) \\ &- \eta J(\theta^{(0)})^\top J(\theta^{(0)})\boldsymbol{Q}^{(t)}\Delta_t \end{aligned} \tag{65}$$

By Lemma 11, we have

$$\begin{aligned} &\left\|J(\tilde{\theta}^{(t)})^\top J(\theta^{(t)}) - J(\theta^{(0)})^\top J(\theta^{(0)})\right\|_F \\ &\leq \left\|\left(J(\tilde{\theta}^{(t)}) - J(\theta^{(0)})\right)^\top J(\theta^{(t)})\right\|_F + \left\|J(\theta^{(0)})^\top\left(J(\theta^{(t)}) - J(\theta^{(0)})\right)\right\|_F \\ &\leq 2M^2 C_0 \tilde{d}^{-1/4} \end{aligned} \tag{66}$$

which implies that for all $t < t_\epsilon$,

$$\begin{aligned} \|\Delta_{t+1}\|_2 &\leq \left\|\left[\boldsymbol{I} - \eta J(\theta^{(0)})^\top J(\theta^{(0)})\boldsymbol{Q}^{(t)}\right]\Delta_t\right\|_2 + \left\|\eta\left[J(\tilde{\theta}^{(t)})^\top J(\theta^{(t)}) - J(\theta^{(0)})^\top J(\theta^{(0)})\right]\boldsymbol{Q}^{(t)}g(\theta^{(t)})\right\|_2 \\ &\leq \left\|\boldsymbol{I} - \eta J(\theta^{(0)})^\top J(\theta^{(0)})\boldsymbol{Q}^{(t)}\right\|_F \|\Delta_t\|_2 + \eta\left\|J(\tilde{\theta}^{(t)})^\top J(\theta^{(t)}) - J(\theta^{(0)})^\top J(\theta^{(0)})\right\|_F \left\|g(\theta^{(t)})\right\|_2 \\ &\leq (1 + \eta M^2)\|\Delta_t\|_2 + 2\eta M^2 C_0 B^t R_0 \tilde{d}^{-1/4} \\ &\leq B\|\Delta_t\|_2 + 2\eta M^2 C_0 B^t R_0 \tilde{d}^{-1/4} \end{aligned} \tag{67}$$

Therefore, we have

$$B^{-(t+1)}\|\Delta_{t+1}\|_2 \leq B^{-t}\|\Delta_t\|_2 + 2\eta M^2 C_0 B^{-1} R_0 \tilde{d}^{-1/4} \tag{68}$$

Since $\Delta_0 = 0$, it follows that for all $t \leq t_\epsilon$,

$$\|\Delta_t\|_2 \leq 2t\eta M^2 C_0 B^{t-1} R_0 \tilde{d}^{-1/4} \tag{69}$$

and particularly we have

$$\left\|\sqrt{\boldsymbol{Q}}\Delta_{t_\epsilon}\right\|_2 \leq \|\Delta_{t_\epsilon}\|_2 \leq 2t_\epsilon\eta M^2 C_0 B^{t_\epsilon-1} R_0 \tilde{d}^{-1/4} \tag{70}$$

For $t \geq t_\epsilon$, we have the alternative update rule (42). Thus,

$$\begin{aligned} \sqrt{\boldsymbol{Q}}\Delta_{t+1} - \sqrt{\boldsymbol{Q}}\Delta_t =& \eta\sqrt{\boldsymbol{Q}}\left[J(\tilde{\theta}^{(t)})^\top J(\theta^{(t)}) - J(\theta^{(0)})^\top J(\theta^{(0)})\right]\sqrt{\boldsymbol{Q}_{3\epsilon}^{(t)}}\left[\sqrt{\boldsymbol{Q}}g(\theta^{(t)})\right] \\ &- \eta\sqrt{\boldsymbol{Q}}J(\theta^{(0)})^\top J(\theta^{(0)})\sqrt{\boldsymbol{Q}_{3\epsilon}^{(t)}}\left[\sqrt{\boldsymbol{Q}}\Delta_t\right] \end{aligned} \tag{71}$$

Let $\boldsymbol{A} = \boldsymbol{I} - \eta\sqrt{\boldsymbol{Q}}J(\theta^{(0)})^\top J(\theta^{(0)})\sqrt{\boldsymbol{Q}_{3\epsilon}^{(t)}} = \boldsymbol{I} - \eta\sqrt{\boldsymbol{Q}}\Theta^{(0)}\sqrt{\boldsymbol{Q}_{3\epsilon}^{(t)}}$. Then, we have

$$\sqrt{\boldsymbol{Q}}\Delta_{t+1} = \boldsymbol{A}\sqrt{\boldsymbol{Q}}\Delta_t + \eta\sqrt{\boldsymbol{Q}}\left[J(\tilde{\theta}^{(t)})^\top J(\theta^{(t)}) - J(\theta^{(0)})^\top J(\theta^{(0)})\right]\sqrt{\boldsymbol{Q}_{3\epsilon}^{(t)}}\left(\sqrt{\boldsymbol{Q}}g(\theta^{(t)})\right) \quad (72)$$

Let $\gamma = 1 - \frac{\eta q^* \lambda^{\min}}{3} < 1$. Combining with Theorem 10 and (62), the above leads to

$$
\begin{aligned}
\left\|\sqrt{\boldsymbol{Q}}\Delta_{t+1}\right\|_2 &\leq \|\boldsymbol{A}\|_2 \left\|\sqrt{\boldsymbol{Q}}\Delta_t\right\|_2 + \eta \left\|\sqrt{\boldsymbol{Q}}\left[J(\tilde{\theta}^{(t)})^\top J(\theta^{(t)}) - J(\theta^{(0)})^\top J(\theta^{(0)})\right]\sqrt{\boldsymbol{Q}_{3\epsilon}^{(t)}}\right\|_2 \left\|\sqrt{\boldsymbol{Q}}g(\theta^{(t)})\right\|_2 \\
&\leq \gamma \left\|\sqrt{\boldsymbol{Q}}\Delta_t\right\|_2 + \eta \left\|J(\tilde{\theta}^{(t)})^\top J(\theta^{(t)}) - J(\theta^{(0)})^\top J(\theta^{(0)})\right\|_F \sqrt{1+3\epsilon}\gamma^{t-t_\epsilon}B^{t_\epsilon}R_0 \\
&\leq \gamma \left\|\sqrt{\boldsymbol{Q}}\Delta_t\right\|_2 + 2\eta M^2 C_0 \sqrt{1+3\epsilon}\gamma^{t-t_\epsilon}B^{t_\epsilon}R_0\tilde{d}^{-1/4}
\end{aligned}
$$
$$(73)$$

This implies that

$$\gamma^{-(t+1)}\left\|\sqrt{\boldsymbol{Q}}\Delta_{t+1}\right\|_2 \leq \gamma^{-t}\left\|\sqrt{\boldsymbol{Q}}\Delta_t\right\|_2 + 2\eta M^2 C_0 \sqrt{1+3\epsilon}\gamma^{-1-t_\epsilon}B^{t_\epsilon}R_0\tilde{d}^{-1/4} \quad (74)$$

Combining with (70), it implies that for all $t \geq t_\epsilon$,

$$\left\|\sqrt{\boldsymbol{Q}}\Delta_t\right\|_2 \leq 2\gamma^{t-t_\epsilon}\eta M^2 C_0 B^{t_\epsilon}R_0\left[t_\epsilon B^{-1} + \sqrt{1+3\epsilon}\gamma^{-1}(t-t_\epsilon)\right]\tilde{d}^{-1/4} \quad (75)$$

Next, we consider an arbitrary test point $\boldsymbol{x}$ such that $\|\boldsymbol{x}\|_2 \leq 1$. Denote $\delta_t = f_{\text{lin}}^{(t)}(\boldsymbol{x}) - f^{(t)}(\boldsymbol{x})$. Then we have

$$
\begin{cases}
f_{\text{lin}}^{(t+1)}(\boldsymbol{x}) - f_{\text{lin}}^{(t)}(\boldsymbol{x}) = -\eta\nabla_\theta f(\boldsymbol{x};\theta^{(0)})^\top J(\theta^{(0)})\boldsymbol{Q}^{(t)}g_{\text{lin}}(\theta^{(t)}) \\
f^{(t+1)}(\boldsymbol{x}) - f^{(t)}(\boldsymbol{x}) = -\eta\nabla_\theta f(\boldsymbol{x};\tilde{\theta}^{(t)})^\top J(\theta^{(t)})\boldsymbol{Q}^{(t)}g(\theta^{(t)})
\end{cases}
\quad (76)
$$

which yields

$$
\begin{aligned}
\delta_{t+1} - \delta_t =& \eta\left[\nabla_\theta f(\boldsymbol{x};\tilde{\theta}^{(t)})^\top J(\theta^{(t)}) - \nabla_\theta f(\boldsymbol{x};\theta^{(0)})^\top J(\theta^{(0)})\right]\boldsymbol{Q}^{(t)}g(\theta^{(t)}) \\
&- \eta\nabla_\theta f(\boldsymbol{x};\theta^{(0)})^\top J(\theta^{(0)})\boldsymbol{Q}^{(t)}\Delta_t
\end{aligned}
\quad (77)
$$

For $t \leq t_\epsilon$, we have

$$
\begin{aligned}
\|\delta_t\|_2 \leq& \eta\sum_{s=0}^{t-1}\left\|\left[\nabla_\theta f(\boldsymbol{x};\tilde{\theta}^{(s)})^\top J(\theta^{(s)}) - \nabla_\theta f(\boldsymbol{x};\theta^{(0)})^\top J(\theta^{(0)})\right]\boldsymbol{Q}^{(s)}\right\|_2 \left\|g(\theta^{(s)})\right\|_2 \\
&+ \eta\sum_{s=0}^{t-1}\left\|\nabla_\theta f(\boldsymbol{x};\theta^{(0)})^\top J(\theta^{(0)})\boldsymbol{Q}^{(s)}\right\|_2 \|\Delta_s\|_2 \\
\leq& \eta\sum_{s=0}^{t-1}\left\|\nabla_\theta f(\boldsymbol{x};\tilde{\theta}^{(s)})^\top J(\theta^{(s)}) - \nabla_\theta f(\boldsymbol{x};\theta^{(0)})^\top J(\theta^{(0)})\right\|_F \left\|g(\theta^{(s)})\right\|_2 \\
&+ \eta\sum_{s=0}^{t-1}\left\|\nabla_\theta f(\boldsymbol{x};\theta^{(0)})\right\|_2 \left\|J(\theta^{(0)})\right\|_F \|\Delta_s\|_2 \\
\leq& 2\eta M^2 C_0 \tilde{d}^{-1/4}\sum_{s=0}^{t-1}B^s R_0 + \eta M^2 \sum_{s=0}^{t-1}(2s\eta M^2 C_0 B^{s-1}R_0\tilde{d}^{-1/4})
\end{aligned}
\quad (78)
$$

So we can see that there exists a constant $C_1$ such that $\|\delta_{t_\epsilon}\|_2 \leq C_1 \tilde{d}^{-1/4}$. Then, for $t > t_\epsilon$, we have

$$
\begin{aligned}
\|\delta_t\|_2 - \|\delta_{t_\epsilon}\|_2 \leq & \eta \sum_{s=t_\epsilon}^{t-1} \left\| \left[ \nabla_\theta f(\boldsymbol{x}; \tilde{\theta}^{(s)})^\top J(\theta^{(s)}) - \nabla_\theta f(\boldsymbol{x}; \theta^{(0)})^\top J(\theta^{(0)}) \right] \sqrt{\boldsymbol{Q}_{3\epsilon}^{(s)}} \right\|_2 \left\| \sqrt{\boldsymbol{Q}} g(\theta^{(s)}) \right\|_2 \\
& + \eta \sum_{s=t_\epsilon}^{t-1} \left\| \nabla_\theta f(\boldsymbol{x}; \theta^{(0)})^\top J(\theta^{(0)}) \sqrt{\boldsymbol{Q}_{3\epsilon}^{(s)}} \right\|_2 \left\| \sqrt{\boldsymbol{Q}} \Delta_s \right\|_2 \\
\leq & 2\eta M^2 C_0 \tilde{d}^{-1/4} \sqrt{1 + 3\epsilon} \sum_{s=t_\epsilon}^{t-1} \gamma^{s-t_\epsilon} B^{t_\epsilon} R_0 \\
& + \eta M^2 \sqrt{1 + 3\epsilon} \sum_{s=t_\epsilon}^{t-1} \left( 2\gamma^{s-t_\epsilon} \eta M^2 C_0 B^{t_\epsilon} R_0 \left[ t_\epsilon B^{-1} + \sqrt{1 + 3\epsilon} \gamma^{-1}(s - t_\epsilon) \right] \tilde{d}^{-1/4} \right)
\end{aligned}
\tag{79}
$$

Note that $\sum_{t=0}^{\infty} t\gamma^t$ is finite as long as $\gamma \in (0, 1)$. Therefore, there is a constant $C$ such that for any $t$, $\|\delta_t\|_2 \leq C\tilde{d}^{-1/4}$ with probability at least $(1 - \delta)$ for any $\tilde{d} \geq \tilde{D}$. $\qquad\square$

### B.4  PROOF OF LEMMA 11

We will use the following theorem regarding the eigenvalues of random Gaussian matrices:

**Theorem 12** (Corollary 5.35 in Vershynin (2010)). *If $\boldsymbol{A} \in \mathbb{R}^{p \times q}$ is a random matrix whose entries are independent standard normal random variables, then for every $t \geq 0$, with probability at least $1 - 2\exp(-t^2/2)$,*

$$
\sqrt{p} - \sqrt{q} - t \leq \lambda^{\min}(\boldsymbol{A}) \leq \lambda^{\max}(\boldsymbol{A}) \leq \sqrt{p} + \sqrt{q} + t
\tag{80}
$$

By this theorem, and also note that $W^L$ is a vector, we can see that for any $\delta$, there exist $\tilde{D} > 0$ and $M_1 > 0$ such that if $\tilde{d} \geq \tilde{D}$, then with probability at least $(1 - \delta)$, for all $\theta \in B(\theta^{(0)}, C_0)$, we have

$$
\left\| W^l \right\|_2 \leq 3\sqrt{\tilde{d}} \quad (\forall 0 \leq l \leq L - 1) \qquad \text{and} \qquad \left\| W^L \right\|_2 \leq C_0 \leq 3\sqrt[4]{\tilde{d}}
\tag{81}
$$

as well as

$$
\left\| \beta \boldsymbol{b}^l \right\|_2 \leq M_1 \sqrt{\tilde{d}} \qquad (\forall l = 0, \cdots, L)
\tag{82}
$$

Now we assume that (81) and (82) are true. Then, for any $\boldsymbol{x}$ such that $\|\boldsymbol{x}\|_2 \leq 1$,

$$
\begin{aligned}
\left\| \boldsymbol{h}^1 \right\|_2 &= \left\| \frac{1}{\sqrt{d_0}} W^0 \boldsymbol{x} + \beta \boldsymbol{b}^0 \right\|_2 \leq \frac{1}{\sqrt{d_0}} \left\| W^0 \right\|_2 \|\boldsymbol{x}\|_2 + \left\| \beta \boldsymbol{b}^0 \right\|_2 \leq \left( \frac{3}{\sqrt{d_0}} + M_1 \right) \sqrt{\tilde{d}} \\
\left\| \boldsymbol{h}^{l+1} \right\|_2 &= \left\| \frac{1}{\sqrt{\tilde{d}}} W^l \boldsymbol{x}^l + \beta \boldsymbol{b}^l \right\|_2 \leq \frac{1}{\sqrt{\tilde{d}}} \left\| W^l \right\|_2 \left\| \boldsymbol{x}^l \right\|_2 + \left\| \beta \boldsymbol{b}^l \right\|_2 \qquad (\forall l \geq 1) \\
\left\| \boldsymbol{x}^l \right\|_2 &= \left\| \sigma(\boldsymbol{h}^l) - \sigma(\boldsymbol{0}^l) + \sigma(\boldsymbol{0}^l) \right\|_2 \leq L_0 \left\| \boldsymbol{h}^l \right\|_2 + \sigma(0)\sqrt{\tilde{d}} \qquad (\forall l \geq 1)
\end{aligned}
\tag{83}
$$

where $L_0$ is the Lipschitz constant of $\sigma$ and $\sigma(\boldsymbol{0}^l) = (\sigma(0), \cdots, \sigma(0)) \in \mathbb{R}^{d_l}$. By induction, there exists an $M_2 > 0$ such that $\left\| \boldsymbol{x}^l \right\|_2 \leq M_2 \sqrt{\tilde{d}}$ and $\left\| \boldsymbol{h}^l \right\|_2 \leq M_2 \sqrt{\tilde{d}}$ for all $l = 1, \cdots, L$.

Denote $\boldsymbol{\alpha}^l = \nabla_{\boldsymbol{h}^l} f(\boldsymbol{x}) = \nabla_{\boldsymbol{h}^l} \boldsymbol{h}^{L+1}$. For all $l = 1, \cdots, L$, we have $\boldsymbol{\alpha}^l = \text{diag}(\dot{\sigma}(\boldsymbol{h}^l)) \frac{W^{l\top}}{\sqrt{\tilde{d}}} \boldsymbol{\alpha}^{l+1}$ where $\dot{\sigma}(x) \leq L_0$ for all $x \in \mathbb{R}$ since $\sigma$ is $L_0$-Lipschitz, $\boldsymbol{\alpha}^{L+1} = 1$ and $\left\| \boldsymbol{\alpha}^L \right\|_2 = \left\| \text{diag}(\dot{\sigma}(\boldsymbol{h}^L)) \frac{W^{L\top}}{\sqrt{\tilde{d}}} \right\|_2 \leq \frac{3}{\sqrt[4]{\tilde{d}}} L_0$. Then, we can easily prove by induction that there exists an $M_3 > 1$ such that $\left\| \boldsymbol{\alpha}^l \right\|_2 \leq M_3 / \sqrt[4]{\tilde{d}}$ for all $l = 1, \cdots, L$ (note that this is not true for $L + 1$ because $\boldsymbol{\alpha}^{L+1} = 1$).

For $l = 0$, $\nabla_{W^0} f(\boldsymbol{x}) = \frac{1}{\sqrt{d_0}} \boldsymbol{x}^0 \boldsymbol{\alpha}^{1\top}$, so $\|\nabla_{W^l} f(\boldsymbol{x})\|_2 \leq \frac{1}{\sqrt{d_0}} \left\| \boldsymbol{x}^0 \right\|_2 \left\| \boldsymbol{\alpha}^1 \right\|_2 \leq \frac{1}{\sqrt{d_0}} M_3 / \sqrt[4]{\tilde{d}}$. And for any $l = 1, \cdots, L$, $\nabla_{W^l} f(\boldsymbol{x}) = \frac{1}{\sqrt{\tilde{d}}} \boldsymbol{x}^l \boldsymbol{\alpha}^{l+1}$, so $\|\nabla_{W^l} f(\boldsymbol{x})\|_2 \leq \frac{1}{\sqrt{\tilde{d}}} \left\| \boldsymbol{x}^l \right\|_2 \left\| \boldsymbol{\alpha}^{l+1} \right\|_2 \leq M_2 M_3$.

(Note that if $M_3 > 1$, then $\left\| \boldsymbol{\alpha}^{L+1} \right\|_2 \leq M_3$; and since $\tilde{d} \geq 1$, there is $\left\| \boldsymbol{\alpha}^l \right\|_2 \leq M_3$ for $l \leq L$.) Moreover, for $l = 0, \cdots, L$, $\nabla_{\boldsymbol{b}^l} f(\boldsymbol{x}) = \beta \boldsymbol{\alpha}^{l+1}$, so $\left\| \nabla_{\boldsymbol{b}^l} f(\boldsymbol{x}) \right\|_2 \leq \beta M_3$. Thus, if (81) and (82) are true, then there exists an $M_4 > 0$, such that $\left\| \nabla_\theta f(\boldsymbol{x}) \right\|_2 \leq M_4/\sqrt{n}$. And since $\left\| \boldsymbol{x}_i \right\|_2 \leq 1$ for all $i$, so $\left\| J(\theta) \right\|_F \leq M_4$.

Next, we consider the difference in $\nabla_\theta f(\boldsymbol{x})$ between $\theta$ and $\tilde{\theta}$. Let $\tilde{f}, \tilde{W}, \tilde{\boldsymbol{b}}, \tilde{\boldsymbol{x}}, \tilde{\boldsymbol{h}}, \tilde{\boldsymbol{\alpha}}$ be the function and the values corresponding to $\tilde{\theta}$. There is

$$
\begin{aligned}
\left\| \boldsymbol{h}^1 - \tilde{\boldsymbol{h}}^1 \right\|_2 &= \left\| \frac{1}{\sqrt{d_0}} (W^0 - \tilde{W}^0) \boldsymbol{x} + \beta (\boldsymbol{b}^0 - \tilde{\boldsymbol{b}}^0) \right\|_2 \\
&\leq \frac{1}{\sqrt{d_0}} \left\| W^0 - \tilde{W}^0 \right\|_2 \left\| \boldsymbol{x} \right\|_2 + \beta \left\| \boldsymbol{b}^0 - \tilde{\boldsymbol{b}}^0 \right\|_2 \leq \left( \frac{1}{\sqrt{d_0}} + \beta \right) \left\| \theta - \tilde{\theta} \right\|_2 \\
\left\| \boldsymbol{h}^{l+1} - \tilde{\boldsymbol{h}}^{l+1} \right\|_2 &= \left\| \frac{1}{\sqrt{\tilde{d}}} W^l (\boldsymbol{x}^l - \tilde{\boldsymbol{x}}^l) + \frac{1}{\sqrt{\tilde{d}}} (W^l - \tilde{W}^l) \tilde{\boldsymbol{x}}^l + \beta (\boldsymbol{b}^l - \tilde{\boldsymbol{b}}^l) \right\|_2 \\
&\leq \frac{1}{\sqrt{\tilde{d}}} \left\| W^l \right\|_2 \left\| \boldsymbol{x}^l - \tilde{\boldsymbol{x}}^l \right\|_2 + \frac{1}{\sqrt{\tilde{d}}} \left\| W^l - \tilde{W}^l \right\|_2 \left\| \tilde{\boldsymbol{x}}^l \right\|_2 + \beta \left\| \boldsymbol{b}^l - \tilde{\boldsymbol{b}}^l \right\|_2 \\
&\leq 3 \left\| \boldsymbol{x}^l - \tilde{\boldsymbol{x}}^l \right\|_2 + (M_2 + \beta) \left\| \theta - \tilde{\theta} \right\|_2 \qquad (\forall l \geq 1) \\
\left\| \boldsymbol{x}^l - \tilde{\boldsymbol{x}}^l \right\|_2 &= \left\| \sigma(\boldsymbol{h}^l) - \sigma(\tilde{\boldsymbol{h}}^l) \right\|_2 \leq L_0 \left\| \boldsymbol{h}^l - \tilde{\boldsymbol{h}}^l \right\|_2 \qquad (\forall l \geq 1)
\end{aligned}
\tag{84}
$$

By induction, there exists an $M_5 > 0$ such that $\left\| \boldsymbol{x}^l - \tilde{\boldsymbol{x}}^l \right\|_2 \leq M_5 \left\| \theta - \tilde{\theta} \right\|_2$ for all $l$.

For $\boldsymbol{\alpha}^l$, we have $\boldsymbol{\alpha}^{L+1} = \tilde{\boldsymbol{\alpha}}^{L+1} = 1$, and for all $l \geq 1$,

$$
\begin{aligned}
\left\| \boldsymbol{\alpha}^l - \tilde{\boldsymbol{\alpha}}^l \right\|_2 &= \left\| \operatorname{diag}(\dot{\sigma}(\boldsymbol{h}^l)) \frac{W^{l\top}}{\sqrt{\tilde{d}}} \boldsymbol{\alpha}^{l+1} - \operatorname{diag}(\dot{\sigma}(\tilde{\boldsymbol{h}}^l)) \frac{\tilde{W}^{l\top}}{\sqrt{\tilde{d}}} \tilde{\boldsymbol{\alpha}}^{l+1} \right\|_2 \\
&\leq \left\| \operatorname{diag}(\dot{\sigma}(\boldsymbol{h}^l)) \frac{W^{l\top}}{\sqrt{\tilde{d}}} (\boldsymbol{\alpha}^{l+1} - \tilde{\boldsymbol{\alpha}}^{l+1}) \right\|_2 + \left\| \operatorname{diag}(\dot{\sigma}(\boldsymbol{h}^l)) \frac{(W^l - \tilde{W}^l)^\top}{\sqrt{\tilde{d}}} \tilde{\boldsymbol{\alpha}}^{l+1} \right\|_2 \\
&\quad + \left\| \operatorname{diag}((\dot{\sigma}(\boldsymbol{h}^l) - \dot{\sigma}(\tilde{\boldsymbol{h}}^l))) \frac{\tilde{W}^{l\top}}{\sqrt{\tilde{d}}} \tilde{\boldsymbol{\alpha}}^{l+1} \right\|_2 \\
&\leq 3 L_0 \left\| \boldsymbol{\alpha}^{l+1} - \tilde{\boldsymbol{\alpha}}^{l+1} \right\|_2 + \left( M_3 L_0 \tilde{d}^{-1/2} + 3 M_3 M_5 L_1 \tilde{d}^{-1/4} \right) \left\| \theta - \tilde{\theta} \right\|_2
\end{aligned}
\tag{85}
$$

where $L_1$ is the Lipschitz constant of $\dot{\sigma}$. Particularly, for $l = L$, though $\tilde{\boldsymbol{\alpha}}^{L+1} = 1$, since $\left\| \tilde{W}^L \right\|_2 \leq 3 \tilde{d}^{1/4}$, (85) is still true. By induction, there exists an $M_6 > 0$ such that $\left\| \boldsymbol{\alpha}^l - \tilde{\boldsymbol{\alpha}}^l \right\|_2 \leq \frac{M_6}{\sqrt[4]{\tilde{d}}} \left\| \theta - \tilde{\theta} \right\|_2$ for all $l \geq 1$ (note that this is also true for $l = L+1$).

Thus, if (81) and (82) are true, then for all $\theta, \tilde{\theta} \in B(\theta^{(0)}, C_0)$, any $\boldsymbol{x}$ such that $\left\| \boldsymbol{x} \right\|_2 \leq 1$, we have

$$
\begin{aligned}
\left\| \nabla_{W^0} f(\boldsymbol{x}) - \nabla_{\tilde{W}^0} \tilde{f}(\boldsymbol{x}) \right\|_2 &= \frac{1}{\sqrt{d_0}} \left\| \boldsymbol{x} \boldsymbol{\alpha}^{1\top} - \boldsymbol{x} \tilde{\boldsymbol{\alpha}}^{1\top} \right\|_2 \\
&\leq \frac{1}{\sqrt{d_0}} \left\| \boldsymbol{\alpha}^1 - \tilde{\boldsymbol{\alpha}}^1 \right\|_2 \\
&\leq \frac{1}{\sqrt{d_0}} \frac{M_6}{\sqrt[4]{\tilde{d}}} \left\| \theta - \tilde{\theta} \right\|_2
\end{aligned}
\tag{86}
$$

and for $l = 1, \cdots, L$, we have

$$
\begin{aligned}
\left\| \nabla_{W^l} f(\boldsymbol{x}) - \nabla_{\tilde{W}^l} \tilde{f}(\boldsymbol{x}) \right\|_2 &= \frac{1}{\sqrt{\tilde{d}}} \left\| \boldsymbol{x}^l \boldsymbol{\alpha}^{l+1\top} - \tilde{\boldsymbol{x}}^l \tilde{\boldsymbol{\alpha}}^{l+1\top} \right\|_2 \\
&\leq \frac{1}{\sqrt{\tilde{d}}} \left( \left\| \boldsymbol{x}^l \right\|_2 \left\| \boldsymbol{\alpha}^{l+1} - \tilde{\boldsymbol{\alpha}}^{l+1} \right\|_2 + \left\| \boldsymbol{x}^l - \tilde{\boldsymbol{x}}^l \right\|_2 \left\| \tilde{\boldsymbol{\alpha}}^{l+1} \right\|_2 \right) \quad (87) \\
&\leq \left( \frac{M_2 M_6}{\sqrt[4]{\tilde{d}}} + \frac{M_5 M_3}{\sqrt{\tilde{d}}} \right) \left\| \theta - \tilde{\theta} \right\|_2
\end{aligned}
$$

Moreover, for any $l = 0, \cdots, L$, there is

$$
\left\| \nabla_{b^l} f(\boldsymbol{x}) - \nabla_{\tilde{b}^l} \tilde{f}(\boldsymbol{x}) \right\|_2 = \beta \left\| \boldsymbol{\alpha}^{l+1} - \tilde{\boldsymbol{\alpha}}^{l+1} \right\|_2 \leq \frac{\beta M_6}{\sqrt[4]{\tilde{d}}} \left\| \theta - \tilde{\theta} \right\|_2 \quad (88)
$$

Overall, we can see that there exists a constant $M_7 > 0$ such that $\left\| \nabla_\theta f(\boldsymbol{x}) - \nabla_{\tilde{\theta}} \tilde{f}(\boldsymbol{x}) \right\|_2 \leq \frac{M_7}{\sqrt{n} \cdot \sqrt[4]{\tilde{d}}} \left\| \theta - \tilde{\theta} \right\|_2$, so that $\left\| J(\theta) - J(\tilde{\theta}) \right\|_F \leq \frac{M_7}{\sqrt[4]{\tilde{d}}} \left\| \theta - \tilde{\theta} \right\|_2$. $\qquad \square$

### B.5 Proof of Theorem 6

Let $\eta_1 = \min\{\eta_0, \eta^*\}$, where $\eta_0$ is defined in Corollary 3 and $\eta^*$ is defined in Theorem 5. Let $f_{\text{lin}}^{(t)}(\boldsymbol{x})$ and $f_{\text{linERM}}^{(t)}(\boldsymbol{x})$ be the linearized neural networks of $f^{(t)}(\boldsymbol{x})$ and $f_{\text{ERM}}^{(t)}(\boldsymbol{x})$, respectively. By Theorem 5, for any $\delta > 0$, there exists $\tilde{D} > 0$ and a constant $C$ such that

$$
\begin{cases}
\sup\limits_{t \geq 0} \left| f_{\text{lin}}^{(t)}(\boldsymbol{x}) - f^{(t)}(\boldsymbol{x}) \right| \leq C \tilde{d}^{-1/4} \\
\sup\limits_{t \geq 0} \left| f_{\text{linERM}}^{(t)}(\boldsymbol{x}) - f_{\text{ERM}}^{(t)}(\boldsymbol{x}) \right| \leq C \tilde{d}^{-1/4}
\end{cases} \quad (89)
$$

By Corollary 3, we have

$$
\lim_{t \to \infty} \left| f_{\text{lin}}^{(t)}(\boldsymbol{x}) - f_{\text{linERM}}^{(t)}(\boldsymbol{x}) \right| = 0 \quad (90)
$$

Summing the above yields

$$
\limsup_{t \to \infty} \left| f^{(t)}(\boldsymbol{x}) - f_{\text{ERM}}^{(t)}(\boldsymbol{x}) \right| \leq 2C \tilde{d}^{-1/4} \quad (91)
$$

which is the result we want. $\qquad \square$

### B.6 Proof of Theorem 7

To minimize the regularized risk (12) with gradient descent, the update rule is

$$
\theta^{(t+1)} = \theta^{(t)} - \eta \sum_{i=1}^n q_i^{(t)} \nabla_\theta \ell(f^{(t)}(\boldsymbol{x}_i), y_i) - \eta \mu (\theta^{(t)} - \theta^{(0)}) \quad (92)
$$

We can see that under the new rule, $\theta^{(t)} - \theta^{(0)} \in \text{span}(\nabla_\theta f^{(0)}(\boldsymbol{x}_1), \cdots, \nabla_\theta f^{(0)}(\boldsymbol{x}_n))$ is still true for all $t$. Let $\theta^*$ be the interpolator in $\text{span}(\nabla_\theta f^{(0)}(\boldsymbol{x}_1), \cdots, \nabla_\theta f^{(0)}(\boldsymbol{x}_n))$, then the empirical risk of $\theta$ is $\frac{1}{2n} \sum_{i=1}^n \langle \theta - \theta^*, \nabla_\theta f^{(0)}(\boldsymbol{x}_i) \rangle^2 = \frac{1}{2n} \left\| \nabla_\theta f^{(0)}(\boldsymbol{X})^\top (\theta - \theta^*) \right\|_2^2$. Thus, there exists $T > 0$ such that for any $t \geq T$,

$$
\left\| \nabla_\theta f^{(0)}(\boldsymbol{X})^\top (\theta^{(t)} - \theta^*) \right\|_2^2 \leq 2n\epsilon \quad (93)
$$

Let the smallest singular value of $\frac{1}{\sqrt{n}} \nabla_\theta f^{(0)}(\boldsymbol{X})$ be $s^{\min}$, and we have $s^{\min} > 0$. Note that the column space of $\nabla_\theta f^{(0)}(\boldsymbol{X})$ is exactly $\text{span}(\nabla_\theta f^{(0)}(\boldsymbol{x}_1), \cdots, \nabla_\theta f^{(0)}(\boldsymbol{x}_n))$. Define $\boldsymbol{H} \in \mathbb{R}^{p \times n}$

such that its columns form an orthonormal basis of this subspace, then there exists $\boldsymbol{G} \in R^{n \times n}$ such that $\nabla_\theta f^{(0)}(\boldsymbol{X}) = \boldsymbol{HG}$, and the smallest singular value of $\frac{1}{\sqrt{n}}\boldsymbol{G}$ is also $s^{\min}$. Since $\theta^{(t)} - \theta^{(0)}$ is also in this subspace, there exists $\boldsymbol{v} \in \mathbb{R}^n$ such that $\theta^{(t)} - \theta^* = \boldsymbol{Hv}$. Then we have $\sqrt{2n\epsilon} \geq \left\|\boldsymbol{G}^\top \boldsymbol{H}^\top \boldsymbol{Hv}\right\|_2 = \left\|\boldsymbol{G}^\top \boldsymbol{v}\right\|_2$. Thus, $\|\boldsymbol{v}\|_2 \leq \frac{\sqrt{2\epsilon}}{s^{\min}}$, which implies

$$\left\|\theta^{(t)} - \theta^*\right\|_2 \leq \frac{\sqrt{2\epsilon}}{s^{\min}} \tag{94}$$

By Corollary 3, if we minimize the unregularized risk with ERM, then $\theta$ always converges to the interpolator $\theta^*$. So for any $t \geq T$ and any test point $\boldsymbol{x}$ such that $\|\boldsymbol{x}\|_2 \leq 1$,

$$|f^{(t)}_{\text{linreg}}(\boldsymbol{x}) - f^{(t)}_{\text{linERM}}(\boldsymbol{x})| = |\langle \theta^{(t)} - \theta^*, \nabla_\theta f^{(0)}(\boldsymbol{x})\rangle| \leq \frac{M_0\sqrt{2\epsilon}}{s^{\min}} \tag{95}$$

which implies (14). $\qquad\square$

### B.7 PROOF OF THEOREM 8

First of all, with some simple linear algebra analysis, we can prove the following proposition:

**Proposition 13.** *For any positive definite symmetric matrix $\boldsymbol{H} \in \mathbb{R}^{n \times n}$, denote its largest and smallest eigenvalues by $\lambda^{\max}$ and $\lambda^{\min}$. Then, for any $\boldsymbol{q} \in \mathbb{R}^n_+$ and $\boldsymbol{Q} = \text{diag}(q_1, \cdots, q_n)$, $\boldsymbol{HQ}$ has $n$ positive eigenvalues that are all in $[\min_i q_i \cdot \lambda^{\min}, \max_i q_i \cdot \lambda^{\max}]$.*

*Proof.* $\boldsymbol{H}$ is a positive definite symmetric matrix, so there exists $\boldsymbol{A} \in \mathbb{R}^{n \times n}$ such that $\boldsymbol{H} = \boldsymbol{A}^\top \boldsymbol{A}$, and $\boldsymbol{A}$ is full-rank. First, any eigenvalue of $\boldsymbol{AQA}^\top$ is also an eigenvalue of $\boldsymbol{A}^\top \boldsymbol{AQ}$ and vice versa, because for any eigenvalue $\lambda$ of $\boldsymbol{AQA}^\top$ we have some $\boldsymbol{v} \neq 0$ such that $\boldsymbol{AQA}^\top \boldsymbol{v} = \lambda \boldsymbol{v}$. Multiplying both sides by $\boldsymbol{A}^\top$ on the left yields $\boldsymbol{A}^\top \boldsymbol{AQ}(\boldsymbol{A}^\top \boldsymbol{v}) = \lambda(\boldsymbol{A}^\top \boldsymbol{v})$ which implies that $\lambda$ is also an eigenvalue of $\boldsymbol{A}^\top \boldsymbol{AQ}$ because $\boldsymbol{A}^\top \boldsymbol{v} \neq 0$ as $\lambda \boldsymbol{v} \neq 0$. We can prove the other direction similarly.

Second, by condition we know that the eigenvalues of $\boldsymbol{A}^\top \boldsymbol{A}$ are all in $[\lambda^{\min}, \lambda^{\max}]$ where $\lambda^{\min} > 0$, which implies for any unit vector $\boldsymbol{v}$, $\boldsymbol{v}^\top \boldsymbol{A}^\top \boldsymbol{Av} \in [\lambda^{\min}, \lambda^{\max}]$, which is equivalent to $\|\boldsymbol{Av}\|_2 \in [\sqrt{\lambda^{\min}}, \sqrt{\lambda^{\max}}]$. Thus, we have $\boldsymbol{v}^\top \boldsymbol{A}^\top \boldsymbol{QAv} \in [\lambda^{\min} \min_i q_i, \lambda^{\max} \max_i q_i]$, which implies that the eigenvalues of $\boldsymbol{A}^\top \boldsymbol{QA}$ are all in $[\lambda^{\min} \min_i q_i, \lambda^{\max} \max_i q_i]$.

Thus, the eigenvalues of $\boldsymbol{HQ} = \boldsymbol{A}^\top \boldsymbol{AQ}$ are all in $[\lambda^{\min} \min_i q_i, \lambda^{\max} \max_i q_i]$. $\qquad\square$

Now return back to the proof of Theorem 8. We still use the shorthand (40). With $L_2$ penalty, the update rule of the reweighting algorithm for the neural network is:

$$\theta^{(t+1)} = \theta^{(t)} - \eta J(\theta^{(t)})\boldsymbol{Q}^{(t)} g(\theta^{(t)}) - \eta\mu(\theta^{(t)} - \theta^{(0)}) \tag{96}$$

And the update rule for the linearized neural network is:

$$\theta^{(t+1)}_{\text{lin}} = \theta^{(t)}_{\text{lin}} - \eta J(\theta^{(0)})\boldsymbol{Q}^{(t)} g_{\text{lin}}(\theta^{(t)}) - \eta\mu(\theta^{(t)}_{\text{lin}} - \theta^{(0)}) \tag{97}$$

First, we need to prove that there exists $D_0$ such that for all $\tilde{d} \geq D_0$, $\sup_{t \geq 0}\left\|\theta^{(t)} - \theta^{(0)}\right\|_2$ is bounded with high probability. Denote $a_t = \theta^{(t)} - \theta^{(0)}$. By (96) we have

$$\begin{aligned} a_{t+1} =& (1 - \eta\mu)a_t - \eta[J(\theta^{(t)}) - J(\theta^{(0)})]\boldsymbol{Q}^{(t)} g(\theta^{(t)}) \\ &- \eta J(\theta^{(0)})\boldsymbol{Q}^{(t)}[g(\theta^{(t)}) - g(\theta^{(0)})] - \eta J(\theta^{(0)})\boldsymbol{Q}^{(t)} g(\theta^{(0)}) \end{aligned} \tag{98}$$

which implies

$$\begin{aligned} \|a_{t+1}\|_2 \leq& \left\|(1 - \eta\mu)\boldsymbol{I} - \eta J(\theta^{(0)})\boldsymbol{Q}^{(t)} J(\tilde{\theta}^{(t)})^\top\right\|_2 \|a_t\|_2 \\ &+ \eta\left\|J(\theta^{(t)}) - J(\theta^{(0)})\right\|_F \left\|g(\theta^{(t)})\right\|_2 + \eta\left\|J(\theta^{(0)})\right\|_F \left\|g(\theta^{(0)})\right\|_2 \end{aligned} \tag{99}$$

where $\tilde{\theta}^{(t)}$ is some linear interpolation between $\theta^{(t)}$ and $\theta^{(0)}$. Our choice of $\eta$ ensures that $\eta\mu < 1$. Similar to (48), we can show that for any $\delta > 0$, there exists a constant $R_0 > 0$ such that with

probability at least $(1 - \delta/3)$, $\left\|g(\theta^{(0)})\right\|_2 < R_0$. Let $M$ be as defined in Lemma 11. Denote $A = \eta M R_0$, and let $C_0 = \frac{4A}{\eta\mu}$ in Lemma 11[6]. By Lemma 11, there exists $D_1$ such that for all $\tilde{d} \geq D_1$, with probability at least $(1 - \delta/3)$, (47) is true.

Now we prove by induction that $\|a_t\|_2 < C_0$. It is true for $t = 0$, so we need to prove that if $\|a_t\|_2 < C_0$, then $\|a_{t+1}\|_2 < C_0$.

For the first term on the right-hand side of (99), we have

$$\left\|(1 - \eta\mu)\boldsymbol{I} - \eta J(\theta^{(0)})\boldsymbol{Q}^{(t)}J(\tilde{\theta}^{(t)})^\top\right\|_2 \leq (1 - \eta\mu)\left\|\boldsymbol{I} - \frac{\eta}{1 - \eta\mu}J(\theta^{(0)})\boldsymbol{Q}^{(t)}J(\theta^{(0)})^\top\right\|_2 \\ + \eta\left\|J(\theta^{(0)})\right\|_F\left\|J(\tilde{\theta}^{(t)}) - J(\theta^{(0)})\right\|_F \quad (100)$$

Like what we have done before, we can show that all non-zero eigenvalues of $J(\theta^{(0)})\boldsymbol{Q}^{(t)}J(\theta^{(0)})^\top$ are eigenvalues of $J(\theta^{(0)})^\top J(\theta^{(0)})\boldsymbol{Q}^{(t)}$. This is because for any $\lambda \neq 0$, if $J(\theta^{(0)})\boldsymbol{Q}^{(t)}J(\theta^{(0)})^\top\boldsymbol{v} = \lambda\boldsymbol{v}$, then $J(\theta^{(0)})^\top J(\theta^{(0)})\boldsymbol{Q}^{(t)}(J(\theta^{(0)})^\top\boldsymbol{v}) = \lambda(J(\theta^{(0)})^\top\boldsymbol{v})$, and $J(\theta^{(0)})^\top\boldsymbol{v} \neq 0$ since $\lambda\boldsymbol{v} \neq 0$, so $\lambda$ is also an eigenvalue of $J(\theta^{(0)})^\top J(\theta^{(0)})\boldsymbol{Q}^{(t)}$. On the other hand, by Theorem 4, $J(\theta^{(0)})^\top J(\theta^{(0)})\boldsymbol{Q}^{(t)}$ converges in probability to $\Theta\boldsymbol{Q}^{(t)}$ whose eigenvalues are all in $[0, \lambda^{\max}]$ by Proposition 13. So there exists $D_2$ such that for all $\tilde{d} \geq D_2$, with probability at least $(1 - \delta/3)$, the eigenvalues of $J(\theta^{(0)})\boldsymbol{Q}^{(t)}J(\theta^{(0)})^\top$ are all in $[0, \lambda^{\max} + \lambda^{\min}]$ for all $t$. Since $\eta/(1 - \eta\mu) \leq (\lambda^{\min} + \lambda^{\max})^{-1}$ by our choice of $\eta$, we have

$$\left\|\boldsymbol{I} - \frac{\eta}{1 - \eta\mu}J(\theta^{(0)})\boldsymbol{Q}^{(t)}J(\theta^{(0)})^\top\right\|_2 \leq 1 \quad (101)$$

On the other hand, we can use (47) since $\|a_t\|_2 < C_0$, so $\left\|J(\theta^{(0)})\right\|_F\left\|J(\tilde{\theta}^{(t)}) - J(\theta^{(0)})\right\|_F \leq \frac{M^2}{\sqrt[4]{\tilde{d}}}C_0$. Therefore, there exists $D_3$ such that for all $\tilde{d} \geq D_3$,

$$\left\|(1 - \eta\mu)\boldsymbol{I} - \eta J(\theta^{(0)})\boldsymbol{Q}^{(t)}J(\tilde{\theta}^{(t)})^\top\right\|_2 \leq 1 - \frac{\eta\mu}{2} \quad (102)$$

For the second term, we have

$$\left\|g(\theta^{(t)})\right\|_2 \leq \left\|g(\theta^{(t)}) - g(\theta^{(0)})\right\|_2 + \left\|g(\theta^{(0)})\right\|_2 \\ \leq \left\|J(\tilde{\theta}^{(t)})\right\|_2\left\|\theta^{(t)} - \theta^{(0)}\right\|_2 + R_0 \leq MC_0 + R_0 \quad (103)$$

And for the third term, we have

$$\eta\left\|J(\theta^{(0)})\right\|_F\left\|g(\theta^{(0)})\right\|_2 \leq \eta M R_0 = A \quad (104)$$

Thus, we have

$$\|a_{t+1}\|_2 \leq \left(1 - \frac{\eta\mu}{2}\right)\|a_t\|_2 + \frac{\eta M(MC_0 + R_0)}{\sqrt[4]{\tilde{d}}} + A \quad (105)$$

So there exists $D_4$ such that for all $\tilde{d} \geq D_4$, $\|a_{t+1}\|_2 \leq \left(1 - \frac{\eta\mu}{2}\right)\|a_t\|_2 + 2A$. This shows that if $\|a_t\|_2 < C_0$ is true, then $\|a_{t+1}\|_2 < C_0$ will also be true.

In conclusion, by union bound, we have proved that for any $\delta > 0$, with probability at least $(1 - \delta)$ for all $\tilde{d} \geq D_0 = \max\{D_1, D_2, D_3, D_4\}$, $\left\|\theta^{(t)} - \theta^{(0)}\right\|_2 < C_0$ is true for all $t$. This also implies that for $C_1 = MC_0 + R_0$, we have $\left\|g(\theta^{(t)})\right\|_2 \leq C_1$ for all $t$ by (103).

---

[6]Note that Lemma 11 only depends on the network structure and does not depend on the update rule, so we can use this lemma here.

Second, let $\Delta_t = \theta_{\text{lin}}^{(t)} - \theta^{(t)}$. Then we have

$$\Delta_{t+1} - \Delta_t = \eta(J(\theta^{(t)})\boldsymbol{Q}^{(t)}g(\theta^{(t)}) - J(\theta^{(0)})\boldsymbol{Q}^{(t)}g_{\text{lin}}(\theta^{(t)}) - \mu\Delta_t) \tag{106}$$

which implies

$$\Delta_{t+1} = \left[(1 - \eta\mu)\boldsymbol{I} - \eta J(\theta^{(0)})\boldsymbol{Q}^{(t)}J(\tilde{\theta}^{(t)})^\top\right]\Delta_t + \eta(J(\theta^{(t)}) - J(\theta^{(0)}))\boldsymbol{Q}^{(t)}g(\theta^{(t)}) \tag{107}$$

By (102), with probability at least $(1 - \delta)$ for all $\tilde{d} \geq D_0$, we have

$$\|\Delta_{t+1}\|_2 \leq \left\|(1 - \eta\mu)\boldsymbol{I} - \eta J(\theta^{(0)})\boldsymbol{Q}^{(t)}J(\tilde{\theta}^{(t)})^\top\right\|_2 \|\Delta_t\|_2 + \eta\left\|J(\theta^{(t)}) - J(\theta^{(0)})\right\|_F \left\|g(\theta^{(t)})\right\|_2$$
$$\leq \left(1 - \frac{\eta\mu}{2}\right)\|\Delta_t\|_2 + \eta\frac{M}{\sqrt[4]{\tilde{d}}}C_0C_1 \tag{108}$$

Again, as $\Delta_0 = 0$, we can prove by induction that for all $t$,

$$\|\Delta_t\|_2 < \frac{2MC_0C_1}{\mu}\tilde{d}^{-1/4} \tag{109}$$

For any test point $\boldsymbol{x}$ such that $\|\boldsymbol{x}\|_2 \leq 1$, we have

$$\begin{aligned}
\left|f_{\text{reg}}^{(t)}(\boldsymbol{x}) - f_{\text{linreg}}^{(t)}(\boldsymbol{x})\right| &= \left|f(\boldsymbol{x};\theta^{(t)}) - f_{\text{lin}}(\boldsymbol{x};\theta_{\text{lin}}^{(t)})\right| \\
&\leq \left|f(\boldsymbol{x};\theta^{(t)}) - f_{\text{lin}}(\boldsymbol{x};\theta^{(t)})\right| + \left|f_{\text{lin}}(\boldsymbol{x};\theta^{(t)}) - f_{\text{lin}}(\boldsymbol{x};\theta_{\text{lin}}^{(t)})\right| \\
&\leq \left|f(\boldsymbol{x};\theta^{(t)}) - f_{\text{lin}}(\boldsymbol{x};\theta^{(t)})\right| + \left\|\nabla_\theta f(\boldsymbol{x};\theta^{(0)})\right\|_2 \left\|\theta^{(t)} - \theta_{\text{lin}}^{(t)}\right\|_2 \\
&\leq \left|f(\boldsymbol{x};\theta^{(t)}) - f_{\text{lin}}(\boldsymbol{x};\theta^{(t)})\right| + M\|\Delta_t\|_2
\end{aligned} \tag{110}$$

For the first term, note that

$$\begin{cases} f(\boldsymbol{x};\theta^{(t)}) - f(\boldsymbol{x};\theta^{(0)}) = \nabla_\theta f(\boldsymbol{x};\tilde{\theta}^{(t)})(\theta^{(t)} - \theta^{(0)}) \\ f_{\text{lin}}(\boldsymbol{x};\theta^{(t)}) - f_{\text{lin}}(\boldsymbol{x};\theta^{(0)}) = \nabla_\theta f(\boldsymbol{x};\theta^{(0)})(\theta^{(t)} - \theta^{(0)}) \end{cases} \tag{111}$$

where $\tilde{\theta}^{(t)}$ is some linear interpolation between $\theta^{(t)}$ and $\theta^{(0)}$. Since $f(\boldsymbol{x};\theta^{(0)}) = f_{\text{lin}}(\boldsymbol{x};\theta^{(0)})$,

$$\left|f(\boldsymbol{x};\theta^{(t)}) - f_{\text{lin}}(\boldsymbol{x};\theta^{(t)})\right| \leq \left\|\nabla_\theta f(\boldsymbol{x};\tilde{\theta}^{(t)}) - \nabla_\theta f(\boldsymbol{x};\theta^{(0)})\right\|_2 \left\|\theta^{(t)} - \theta^{(0)}\right\|_2 \leq \frac{M}{\sqrt[4]{\tilde{d}}}C_0^2 \tag{112}$$

Thus, we have shown that for all $\tilde{d} \geq D_0$, with probability at least $(1 - \delta)$ for all $t$ and all $\boldsymbol{x}$,

$$\left|f_{\text{reg}}^{(t)}(\boldsymbol{x}) - f_{\text{linreg}}^{(t)}(\boldsymbol{x})\right| \leq \left(MC_0^2 + \frac{2M^2C_0C_1}{\mu}\right)\tilde{d}^{-1/4} = O(\tilde{d}^{-1/4}) \tag{113}$$

Given that $\hat{\mathcal{R}}(f_{\text{linreg}}^{(t)}) < \epsilon$ for sufficiently large $t$, this also implies that

$$\left|\hat{\mathcal{R}}(f_{\text{linreg}}^{(t)}) - \hat{\mathcal{R}}(f_{\text{reg}}^{(t)})\right| = O(\tilde{d}^{-1/4}\sqrt{\epsilon} + \tilde{d}^{-1/2}) \tag{114}$$

So for a fixed $\epsilon$, there exists $D > 0$ such that for all $d \geq D$, for sufficiently large $t$,

$$\hat{\mathcal{R}}(f_{\text{reg}}^{(t)}) < \epsilon \Rightarrow \hat{\mathcal{R}}(f_{\text{linreg}}^{(t)}) < 2\epsilon \tag{115}$$

By Theorem 5, we have

$$\sup_{t \geq 0} \left|f_{\text{linERM}}^{(t)}(\boldsymbol{x}) - f_{\text{ERM}}^{(t)}(\boldsymbol{x})\right| = O(\tilde{d}^{-1/4}) \tag{116}$$

Combining Theorem 7 with (113) and (116) derives

$$\limsup_{t \to \infty} \left|f_{\text{reg}}^{(t)}(\boldsymbol{x}) - f_{\text{ERM}}^{(t)}(\boldsymbol{x})\right| = O(\tilde{d}^{-1/4} + \sqrt{\epsilon}) \tag{117}$$

Letting $\tilde{d} \to \infty$ leads to the result we need. $\qquad\square$

## C    A NOTE ON THE PROOFS IN LEE ET AL. (2019)

We have mentioned that the proofs in Lee et al. (2019), particularly the proofs of their Theorem 2.1 and Lemma 1 in their Appendix G, are flawed. In order to fix their proof, we change the network initialization to (9). In this section, we will demonstrate what goes wrong in the proofs in Lee et al. (2019), and how we manage to fix the proof. For clarity, we are referring to the following version of the paper: `https://arxiv.org/pdf/1902.06720v4.pdf`.

To avoid confusion, in this section we will still use the notations used in our paper.

### C.1    THEIR PROBLEMS

Lee et al. (2019) claimed in their Theorem 2.1 that under the conditions of our Theorem 5, for any $\delta > 0$, there exist $\tilde{D} > 0$ and a constant $C$ such that for any $\tilde{d} \geq \tilde{D}$, with probability at least $(1 - \delta)$, the gap between the output of a sufficiently wide fully-connected neural network and the output of its linearized neural network at any test point $\boldsymbol{x}$ can be uniformly bounded by

$$\sup_{t \geq 0} \left| f^{(t)}(\boldsymbol{x}) - f_{\text{lin}}^{(t)}(\boldsymbol{x}) \right| \leq C \tilde{d}^{-1/2} \qquad \text{(claimed)} \tag{118}$$

where they used the original NTK formulation and initialization in Jacot et al. (2018):

$$\begin{cases} \boldsymbol{h}^{l+1} = \dfrac{W^l}{\sqrt{d_l}} \boldsymbol{x}^l + \beta \boldsymbol{b}^l \\ \boldsymbol{x}^{l+1} = \sigma(\boldsymbol{h}^{l+1}) \end{cases} \quad \text{and} \quad \begin{cases} W_{i,j}^{l(0)} \sim \mathcal{N}(0, 1) \\ b_i^{l(0)} \sim \mathcal{N}(0, 1) \end{cases} \quad (\forall l = 0, \cdots, L) \tag{119}$$

where $\boldsymbol{x}_0 = \boldsymbol{x}$ and $f(\boldsymbol{x}) = h^{L+1}$. However, in their proof in their Appendix G, they did not directly prove their result for the NTK formulation, but instead they proved another result for the following formulation which they called the *standard formulation*:

$$\begin{cases} \boldsymbol{h}^{l+1} = W^l \boldsymbol{x}^l + \beta \boldsymbol{b}^l \\ \boldsymbol{x}^{l+1} = \sigma(\boldsymbol{h}^{l+1}) \end{cases} \quad \text{and} \quad \begin{cases} W_{i,j}^{l(0)} \sim \mathcal{N}(0, \dfrac{1}{d_l}) \\ b_i^{l(0)} \sim \mathcal{N}(0, 1) \end{cases} \quad (\forall l = 0, \cdots, L) \tag{120}$$

See their Appendix F for the definition of their standard formulation. In the original formulation, they also included two constants $\sigma_w$ and $\sigma_b$ for standard deviations, and for simplicity we omit these constants here. Note that the outputs of the NTK formulation and the standard formulation at initialization are actually the same. The only difference is that the norm of the weight $W^l$ and the gradient of the model output with respect to $W^l$ are different for all $l$.

In their Appendix G, they claimed that if a network with the standard formulation is trained by minimizing the squared loss with gradient descent and learning rate $\eta' = \eta/\tilde{d}$, where $\eta$ is our learning rate in Theorem 5 and also their learning rate in their Theorem 2.1, then (118) is true for this network, so it is also true for a network with the NTK formulation because the two formulations have the same network output. And then they claimed in their equation (S37) that applying learning rate $\eta'$ to the standard formulation is equivalent to applying the following learning rates

$$\eta_W^l = \frac{d_l}{d_{\max}} \eta \qquad \text{and} \qquad \eta_{\boldsymbol{b}}^l = \frac{1}{d_{\max}} \eta \tag{121}$$

to $W^l$ and $\boldsymbol{b}^l$ of the NTK formulation, where $d_{\max} = \max\{d_0, \cdots, d_L\}$.

To avoid confusion, in the following discussions we will still use the NTK formulation and initialization if not stated otherwise.

**Problem 1.**    Claim (121) is true, but it leads to two problems. The first problem is that $\eta_{\boldsymbol{b}}^l = O(d_{\max}^{-1})$ since $\eta = O(1)$, while their Theorem 2.1 needs the learning rate to be $O(1)$. Nevertheless, this problem can be simply fixed by modifying their standard formulation as $\boldsymbol{h}^{l+1} = W^l \boldsymbol{x}^l + \beta \sqrt{d_l} \boldsymbol{b}^l$ where $b_i^{l(0)} \sim \mathcal{N}(0, d_l^{-1})$. The real problem that is non-trivial to fix is that by (121), there is

$\eta_W^0 = \frac{d_0}{d_{\max}}\eta$. However, note that $d_0$ is a constant since it is the dimension of the input space, while $d_{\max}$ goes to infinity. With that being said, in (121) they were essentially using a very small learning rate for the first layer $W^0$ but a normal learning rate for the rest of the layers, which definitely does not match with their claim in their Theorem 2.1.

**Problem 2.** Another big problem is that the proof of their Lemma 1 in their Appendix G is erroneous, and consequently their Theorem 2.1 is unsound as it heavily depends on their Lemma 1. In their Lemma 1, they claimed that for some constant $M > 0$, for any two models with the parameters $\theta$ and $\tilde{\theta}$ such that $\theta, \tilde{\theta} \in B(\theta^{(0)}, C_0)$ for some constant $C_0$, there is

$$\left\| J(\theta) - J(\tilde{\theta}) \right\|_F \leq \frac{M}{\sqrt{\tilde{d}}} \left\| \theta - \tilde{\theta} \right\|_2 \qquad \text{(claimed)} \qquad (122)$$

Note that the original claim in their paper was $\left\| J(\theta) - J(\tilde{\theta}) \right\|_F \leq M\sqrt{\tilde{d}} \left\| \theta - \tilde{\theta} \right\|_2$. This is because they were proving this result for their standard formulation. Compared to the standard formulation, in the NTK formulation $\theta$ is $\sqrt{\tilde{d}}$ times larger, while the Jacobian $J(\theta)$ is $\sqrt{\tilde{d}}$ times smaller. This is also why here we have $\theta, \tilde{\theta} \in B(\theta^{(0)}, C_0)$ instead of $\theta, \tilde{\theta} \in B(\theta^{(0)}, C_0\tilde{d}^{-1/2})$ for the NTK formulation. Therefore, equivalently they were claiming (122) for the NTK formulation.

However, their proof of (122) in incorrect. Specifically, the right-hand side of their inequality (S86) is incorrect. Using the notations in our Appendix B.4, their (S86) essentially claimed that

$$\left\| \boldsymbol{\alpha}^l - \tilde{\boldsymbol{\alpha}}^l \right\|_2 \leq \frac{M}{\sqrt{\tilde{d}}} \left\| \theta - \tilde{\theta} \right\|_2 \qquad \text{(claimed)} \qquad (123)$$

for any $\theta, \tilde{\theta} \in B(\theta^{(0)}, C_0)$, where $\boldsymbol{\alpha}^l = \nabla_{\boldsymbol{h}^l} \boldsymbol{h}^{L+1}$ and $\tilde{\boldsymbol{\alpha}}^l$ is the same gradient for the second model. Note that their (S86) does not have the $\sqrt{\tilde{d}}$ in the denominator which appears in (123). This is because for their standard formulation, $\theta$ is $\sqrt{\tilde{d}}$ times smaller than the original NTK formulation, while $\left\| \boldsymbol{\alpha}^l \right\|_2$ has the same order in the two formulations because all $\boldsymbol{h}^l$ are the same.

However, it is actually impossible to prove (123). Consider the following counterexample: Since $\theta$ and $\tilde{\theta}$ are arbitrarily chosen, we can choose them such that they only differ in $b_1^l$ for some $1 \leq l < L$. Then, $\left\| \theta - \tilde{\theta} \right\|_2 = \left| b_1^l - \tilde{b}_1^l \right|$. We can see that $\boldsymbol{h}^{l+1}$ and $\tilde{\boldsymbol{h}}^{l+1}$ only differ in the first element, and $\left| h_1^{l+1} - \tilde{h}_1^{l+1} \right| = \left| \beta(b_1^l - \tilde{b}_1^l) \right|$. Moreover, we have $W^{l+1} = \tilde{W}^{l+1}$, so there is

$$\begin{aligned}
\boldsymbol{\alpha}^{l+1} - \tilde{\boldsymbol{\alpha}}^{l+1} &= \text{diag}(\dot{\sigma}(\boldsymbol{h}^{l+1}))\frac{W^{l+1\top}}{\sqrt{\tilde{d}}}\boldsymbol{\alpha}^{l+2} - \text{diag}(\dot{\sigma}(\tilde{\boldsymbol{h}}^{l+1}))\frac{\tilde{W}^{l+1\top}}{\sqrt{\tilde{d}}}\tilde{\boldsymbol{\alpha}}^{l+2} \\
&= \left[ \text{diag}(\dot{\sigma}(\boldsymbol{h}^{l+1})) - \text{diag}(\dot{\sigma}(\tilde{\boldsymbol{h}}^{l+1})) \right] \frac{W^{l+1\top}}{\sqrt{\tilde{d}}}\boldsymbol{\alpha}^{l+2} \\
&\quad + \text{diag}(\dot{\sigma}(\tilde{\boldsymbol{h}}^{l+1}))\frac{W^{l+1\top}}{\sqrt{\tilde{d}}}(\boldsymbol{\alpha}^{l+2} - \tilde{\boldsymbol{\alpha}}^{l+2})
\end{aligned} \qquad (124)$$

Then we can lower bound $\left\| \boldsymbol{\alpha}^{l+1} - \tilde{\boldsymbol{\alpha}}^{l+1} \right\|_2$ by

$$\begin{aligned}
\left\| \boldsymbol{\alpha}^{l+1} - \tilde{\boldsymbol{\alpha}}^{l+1} \right\|_2 &\geq \left\| \left[ \text{diag}(\dot{\sigma}(\boldsymbol{h}^{l+1})) - \text{diag}(\dot{\sigma}(\tilde{\boldsymbol{h}}^{l+1})) \right] \frac{W^{l+1\top}}{\sqrt{\tilde{d}}}\boldsymbol{\alpha}^{l+2} \right\|_2 \\
&\quad - \left\| \text{diag}(\dot{\sigma}(\tilde{\boldsymbol{h}}^{l+1}))\frac{W^{l+1\top}}{\sqrt{\tilde{d}}}(\boldsymbol{\alpha}^{l+2} - \tilde{\boldsymbol{\alpha}}^{l+2}) \right\|_2
\end{aligned} \qquad (125)$$

The first term on the right-hand side is equal to $\left| \left[ \dot{\sigma}(h_1^{l+1}) - \dot{\sigma}(\tilde{h}_1^{l+1}) \right] \langle W_1^{l+1}/\sqrt{\tilde{d}}, \boldsymbol{\alpha}^{l+2} \rangle \right|$ where $W_1^{l+1}$ is the first row of $W^{l+1}$. We know that $\left\| W_1^{l+1} \right\|_2 = \Theta\left( \sqrt{\tilde{d}} \right)$ with high probability as its

elements are sampled from $\mathcal{N}(0, 1)$, and in their (S85) they claimed that $\left\|\boldsymbol{\alpha}^{l+2}\right\|_2 = O(1)$, which is true. In addition, they assumed that $\dot{\sigma}$ is Lipschitz. Hence, we can see that

$$\left\|\left[\operatorname{diag}(\dot{\sigma}(\boldsymbol{h}^{l+1})) - \operatorname{diag}(\dot{\sigma}(\tilde{\boldsymbol{h}}^{l+1}))\right] \frac{W^{l+1\top}}{\sqrt{\tilde{d}}} \boldsymbol{\alpha}^{l+2}\right\|_2 = O\left(\left|h_1^{l+1} - \tilde{h}_1^{l+1}\right|\right) = O\left(\left\|\theta - \tilde{\theta}\right\|_2\right)$$
(126)

On the other hand, suppose that claim (123) is true, then $\left\|\boldsymbol{\alpha}^{l+2} - \tilde{\boldsymbol{\alpha}}^{l+2}\right\|_2 = O\left(\tilde{d}^{-1/2}\left\|\theta - \tilde{\theta}\right\|_2\right)$. Then we can see that the second term on the right-hand side is $O\left(\tilde{d}^{-1/2}\left\|\theta - \tilde{\theta}\right\|_2\right)$ because $\left\|W^{l+1}\right\|_2 = O(\sqrt{\tilde{d}})$ and $\dot{\sigma}(x)$ is bounded by a constant as $\sigma$ is Lipschitz. Thus, for a very large $\tilde{d}$, the second-term is an infinitely small term compared to the first term, so we can only prove that

$$\left\|\boldsymbol{\alpha}^{l+1} - \tilde{\boldsymbol{\alpha}}^{l+1}\right\|_2 = O\left(\left\|\theta - \tilde{\theta}\right\|_2\right)$$
(127)

which is different from (123) because it lacks a critical $\tilde{d}^{-1/2}$ and thus leads to a contradiction. Hence, we cannot prove (123) with the $\tilde{d}^{-1/2}$ factor, and consequently we cannot prove (122) with the $\sqrt{\tilde{d}}$ in the denominator on the right-hand side. As a result, their Lemma 1 and Theorem 2.1 cannot be proved without this critical $\tilde{d}^{-1/2}$. Similarly, we can also construct a counterexample where $\theta$ and $\tilde{\theta}$ only differ in the first row of some $W^l$.

## C.2 OUR FIXES

Regarding Problem 1, we can still use an $O(1)$ learning rate for the first layer in the NTK formulation given that $\|\boldsymbol{x}\|_2 \leq 1$. This is because for the first layer, we have

$$\nabla_{W^0} f(\boldsymbol{x}) = \frac{1}{\sqrt{d_0}} \boldsymbol{x}^0 \boldsymbol{\alpha}^{1\top} = \frac{1}{\sqrt{d_0}} \boldsymbol{x} \boldsymbol{\alpha}^{1\top}$$
(128)

For all $l \geq 1$, we have $\left\|\boldsymbol{x}^l\right\|_2 = O(\tilde{d}^{1/2})$. However, for $l = 0$, we instead have $\left\|\boldsymbol{x}^0\right\|_2 = O(1)$. Thus, we can prove that the norm of $\nabla_{W^0} f(\boldsymbol{x})$ has the same order as the gradient with respect to any other layer, so there is no need to use a smaller learning rate for the first layer.

Regarding Problem 2, in our formulation (8) and initialization (9), the initialization of the last layer of the NTK formulation is changed from the Gaussian initialization $W_{i,j}^{L(0)} \sim \mathcal{N}(0, 1)$ to the zero initialization $W_{i,j}^{L(0)} = 0$. Now we show how this modification solves Problem 2.

The main consequence of changing the initialization of the last layer is that (81) becomes different: instead of $\left\|W^L\right\|_2 \leq 3\sqrt{\tilde{d}}$, we now have $\left\|W^L\right\|_2 \leq C_0 \leq 3\sqrt[4]{\tilde{d}}$. In fact, for any $r \in (0, 1/2)$, we can prove that $\left\|W^L\right\|_2 \leq 3\tilde{d}^r$ for sufficiently large $\tilde{d}$. In our proof we choose $r = 1/4$.

Consequently, instead of $\left\|\boldsymbol{\alpha}^l\right\|_2 \leq M_3$, we can now prove that $\left\|\boldsymbol{\alpha}^l\right\|_2 \leq M_3 \tilde{d}^{r-1/2}$ for all $l \leq L$ by induction. So now we can prove $\left\|\boldsymbol{\alpha}^l - \tilde{\boldsymbol{\alpha}}^l\right\|_2 = O\left(\tilde{d}^{r-1/2}\left\|\theta - \tilde{\theta}\right\|_2\right)$ instead of $O\left(\left\|\theta - \tilde{\theta}\right\|_2\right)$, because

- For $l < L$, we now have $\left\|\boldsymbol{\alpha}^{l+1}\right\|_2 = O(\tilde{d}^{r-1/2})$ instead of $O(1)$, so we can have the additional $\tilde{d}^{r-1/2}$ factor in the bound.
- For $l = L$, although $\left\|\boldsymbol{\alpha}^{L+1}\right\|_2 = 1$, note that $\left\|W^L\right\|_2$ now becomes $O(\tilde{d}^r)$ instead of $O(\tilde{d}^{1/2})$, so again we can decrease the bound by a factor of $\tilde{d}^{r-1/2}$.

Then, with this critical $\tilde{d}^{r-1/2}$, we can prove the approximation theorem with the form

$$\sup_{t \geq 0} \left|f^{(t)}(\boldsymbol{x}) - f_{\text{lin}}^{(t)}(\boldsymbol{x})\right| \leq C\tilde{d}^{r-1/2}$$
(129)

for any $r \in (0, 1/2)$, though we cannot really prove the $O(\tilde{d}^{-1/2})$ bound as originally claimed in (118). So this is how we solve Problem 2.

One caveat of changing the initialization to zero initialization is whether we can still safely assume that $\lambda^{\min} > 0$ where $\lambda^{\min}$ is the smallest eigenvalue of $\Theta$, the kernel matrix of our new formulation. The answer is yes. In fact, in our Theorem 4 we proved that $\Theta$ is non-degenerated (which means that $\Theta(\boldsymbol{x}, \boldsymbol{x}')$ still depends on $\boldsymbol{x}$ and $\boldsymbol{x}'$), and under the overparameterized setting where $d_L \gg n$, chances are high that $\Theta$ is full-rank. Hence, we can still assume that $\lambda^{\min} > 0$.

As a final remark, one key reason why we need to initialize $W^L$ as zero is that the dimension of the output space (i.e. the dimension of $\boldsymbol{h}^{L+1}$) is finite, and in our case it is 1. Suppose we allow the dimension of $\boldsymbol{h}^{L+1}$ to be $\tilde{d}$ which goes to infinity, then using the same proof techniques, for the NTK formulation we can prove that $\sup_t \left\| \boldsymbol{h}^{L+1(t)} - \boldsymbol{h}_{\mathrm{lin}}^{L+1(t)} \right\|_2 \leq C$, i.e. the gap between two vectors of infinite dimension is always bounded by a finite constant. This is the approximation theorem we need for the infinite-dimensional output space. However, when the dimension of the output space is finite, $\sup_t \left\| \boldsymbol{h}^{L+1(t)} - \boldsymbol{h}_{\mathrm{lin}}^{L+1(t)} \right\|_2 \leq C$ no longer suffices, so we need to decrease the order of the norm of $W^L$ in order to obtain a smaller bound.

## D  EXPERIMENT DETAILS AND ADDITIONAL EXPERIMENTS

### D.1  EXPERIMENT DETAILS

All experiments are conducted on a Ubuntu 18.04.6 machine with NVIDIA Geforce GTX 1080ti GPUs. Each model is trained with one GPU. On each of Waterbirds and CelebA, we use a ResNet18 as the model. The model is trained with SGD with momentum = 0.9. On Waterbirds the learning rate is $10^{-4}$, and on CelebA it is $10^{-3}$. For Group DRO, $\nu$ is selected as 0.01 (see the definition of $\nu$ in (3)). The batch size used for Waterbirds is 128, and for CelebA it is 400. Data augmentation including random cropping, random horizontal flip and normalization is performed on both datasets.

### D.2  SAMPLE WEIGHTS CONVERGE IN GROUP DRO

The results in Section 3 require Assumption 1 which states that each sample weight $q_i^{(t)}$ converges to some positive value as $t \to \infty$. Our readers might wonder how strong this assumption is, and whether reweighting algorithms satisfy this assumption in practice. In this section we empirically demonstrate that for Group DRO, the dynamic reweighting algorithm we experiment on, this assumption is satisfied on Waterbirds and CelebA.

Recall that in Section 2.2 we empirically showed that reweighting algorithms could easily overfit without regularization. Here using the same experimental settings, we keep track of the weight of each group $g_k$ during training, and we plot the group weight curves in Figure 3. We also train the models longer (1000 epochs on Waterbirds and 300 epochs on CelebA). Clearly we can see that as the training accuracy converges to 100%, the group weights also converge to an equilibrium. Note that $q_i^{(t)} = g_k^{(t)}/n_k$ for all $z_i \in \mathcal{D}_k$, so the sample weights also converge.

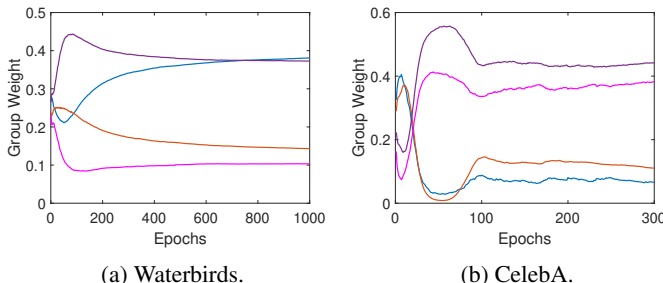

(a) Waterbirds.                    (b) CelebA.

Figure 3: Weights of each group in Group DRO on Waterbirds and CelebA. The four curves correspond to the four groups.

