# OpenReview forum: "Understanding Overfitting in Reweighting Algorithms for Worst-group Performance"
_ICLR.cc/2022/Conference — ICLR 2022 Submitted_

### Official Review · Reviewer_fZgx · 2021-11-02

**Correctness:** 3
**Technical Novelty And Significance:** 3
**Empirical Novelty And Significance:** 2
**Recommendation:** 3
**Confidence:** 3

**Main Review:**

Strengths:
1.	The paper provides good proof for linear models via the intuition that the change in model parameters always lies in a low-dimensional subspace.
2.	The conclusion gives new insight for designing algorithms, which shows reweighting algorithms without any repair methods may not be better than ERM for the worst-group performance.
3.	The paper writing is generally of high quality.


Weaknesses:
1.	My feeling is that the conclusion is somewhat overclaimed. In both abstract and conclusion, it is emphasized that this work proves the pessimistic result that reweighting algorithms always overfit. However, the paper only proves that this conclusion might be true for some **specific** situations. For example, the reweighting algorithms need to satisfy Assumption 1 and Assumption 2, which means not all reweighting algorithms are considered. The overparameterized models need to be linear models, linearized neural networks or wide fully-connected neural networks, which are not commonly used in practice. Besides, the squared loss needs to be used to confirm the update rule is linear. All those assumptions are not quite mild for me.
2.	The analysis of neural networks contributes less. With the existing NTK theorem, the extension from linear models to wide fully-connected neural networks is trivial (Section 3.2, 3.3). The work bypasses the core problem of overparametrized neural networks and only considers the easy wide fully-connected neural networks.
3.	The theoretical results and experiments do not match. The theoretical proof considers wide fully-connected neural networks, while the experiments utilize a ResNet18 as the model, which is quite different.
4.	Some key steps are empirical, although the paper claims that it provides a theoretical backing in the abstract. For example, this paper only proves that reweighting algorithms will converge to the same level as ERM, but the conclusion that ERM has a poor worst-group test performance is summarized through observation in practice. Besides, the paper can only empirically demonstrate that commonly used algorithms satisfy Assumption 2.


**Summary Of The Paper:**

This paper mainly focuses on the reweighting algorithms (e.g. Importance Weighting, Group DRO) for the worst-group performance. It tries to theoretically explain why overfitting problems always appears in reweighting algorithms. Specifically, the authors prove that under several conditions (e.g. assumptions for algorithms, wide fully-connected neural networks, squared loss), the worst-group test performance of the reweighting algorithm will converge to the same level as ERM.

**Summary Of The Review:**

The actual main contribution of this paper is proving that for linear overparameterized models, the current reweighting algorithms almost always overfit. It provides a possible way of analyzing overfitting problems by reducing a family of algorithms to the commonly used ERM. However, the analysis of neural networks contributes less, and some key steps are still empirical results. Besides, there is a gap between theoretical results and experiments. In summary, the conclusion of the paper is somewhat overclaimed, and there is still a lot of room for improvement.

---

> ### Author Response · Authors · 2021-11-22
> **Response to Reviewer fZgx**
>
> We thank the reviewer for the detailed and useful comments. We would like to address your concerns as follows:
>
> 1. We agree that the abstract and the introduction part can be misleading. In the new version of this paper, we will rephrase the statements in the abstract, and clearly state our assumptions in the introduction.
>
> 2. Though our results are still proved for "specific situations", we have made them as general as possible. Previous results are all limited to linear models and static reweighting algorithms, and we consider neural networks and dynamic reweighting algorithms. We will discuss this improvement over previous results in the introduction in the new version.
>
> 3. We strongly disagree that the transfer of results from the linear model to the sufficiently wide neural network is "trivial", for the following two reasons:
>
> (a) The famous (claimed) NTK result in [1] is wrong, and we fix their result and proofs (See Appendix C), which is of course non-trivial.
>
> (b) The previous result only considers ERM, while we prove for dynamic reweighting algorithms, and dynamic reweighting algorithms with regularization. Extending the previous result to static reweighting algorithms is indeed easy (assuming that the previous result is correct, which is unfortunately not true), but it is by no means easy for dynamic ones, let alone dynamic + regularization.
>
> Extending the results to sufficiently wide neural networks using NTK seems intuitive, but technically it is by no means "trivial". If the reviewer is interested in the proof techniques, please read Appendix B though it is a bit long.
>
> 4. "ERM has a poor worst-group test performance" is not a valid theoretical statement and can only be empirically observed, because it depends on the group allocation in a specific task. For example, if there is only one group which is the entire data domain, then the worst-group test performance becomes the average test performance, and ERM will have a very high "worst-group test performance". On the other hand, "ERM has a poor worst-group test performance" is widely observed in practice, and in fact one of the main research goals in the area of fairness is to do better than ERM, so it is implicitly assumed that ERM cannot lead to fair models in this area. If ERM itself leads to fair models then there is even no need for this area to exist.
>
> 5. Regarding the assumptions: We do our best to make our settings as general as possible, though necessary assumptions are still needed. As mentioned in the general response, we have weakened Assumptions 1 and 2 to the new Assumption 1. The assumptions we make can be verified empirically, and theoretically this is already a great improvement from previous results that are limited to linear models and static reweighting algorithms. We will clearly state our assumptions in the introduction in the new version.
>
> We hope that our response addresses your concerns and would like to answer any further questions.
>
> [1] Lee et al., Wide neural networks of any depth evolve as linear models under gradient descent. Advances in neural information processing systems, 32:8572–8583, 2019.

---

### Official Review · Reviewer_M2hT · 2021-11-02

**Correctness:** 3
**Technical Novelty And Significance:** 3
**Empirical Novelty And Significance:** 2
**Recommendation:** 6
**Confidence:** 3

**Main Review:**


I enjoyed reading this paper. It is written in a very clear way and it addresses a very timely and important problem. I also enjoyed that it combined both theoretical results and simulations.

Below are some questions that I have and some suggestions that perhaps can be helpful to the authors in improving their paper.

1) If I understand correctly from equation (4) and Assumption 1, the framework of the present paper only applies to training updates based on the full dataset, and not to batch or stochastic updates. If this is the case, then this should be highlighted more prominently in the text and in the abstract. Would it be possible to generalize the analysis?

2) "Each $\mathcal{D}_i$ is a subset of $\mathcal{X} \times \mathcal{Y}$": Is this condition required for the theoretical results? If so, it should be discussed at more length. Can it be relaxed? For example, in my mind often the notion of worst-case performance relates to multiple environmentsl as in e.g, the conceptual framework of Rothenhäusler et al (2021, JRSSB) or to protected attributes that are not directly included in the model. These may not be perfectly recoverable through subsets of predictors and response variables (though of course they may be strongly correlated).

3) Maybe in the high-level statement after Theorem 2 one could clarify that this statement depends on identical initialization $\theta^{(0)}$ for all involved methods?

4) Last paragraph of Section 3, "Moreover our theoretical results can explain the surprising empirical observation": I think this paragraph is not sufficiently justified (i.e., it does not really explain the observation, at most it hints at the plausibility of it). Can the authors make a stronger case for their argument? Otherwise my suggestion would be soften the claim (or even remove the paragraph).

5) I like the experiments since they bring the main point of the paper across very clearly. One additional result I would have liked to see is an empirical illustration of whether the theoretical arguments hold, e.g., are the final solutions of ERM, DRO, and IW initialized at the same $\theta^{(0)}$ very similar (and more similar than solutions of the same method initialized at different $\theta^{(0)}$)? By similarity here I refer to similarity of actual model fits, rather than accuracy/performance.



# Typographical errors

* Equation (5), $f^{(t)}$ instead of  $f$
* Page 8: "trained trained" (repeated twice)
* Reproducibility statement: I think speculation should be "specification" (note: the word speculation appears twice in this sentence). Anaconda should be capitalized.



**Summary Of The Paper:**

This work studies the problem of training for worst-case subgroup performance. This has been a very prominent research avenue in the past few years and provides a natural approach towards learning fair and "causal" models (that do not perform too badly for any pre-specified subgroup). In practice however there have been some difficulties with existing approaches, and e.g., in the overparameterization regime, these methods do not perform well in terms of  worst-case performance on test sets. This paper provides a theoretical explanation for this fact (using linear models and linearized neural networks) and also illustrates it using numerical experiments on two datasets. Furthermore, the authors assess the role of $L_2$ regularization in training. The practical takeaway is that existing approaches based on reweighting require substantial regularization or early stopping to perform well in terms of test accuracy (in the overparamerized regime).


**Summary Of The Review:**


I think this is an interesting contribution on the theoretical and practical study of worst-case subgroup performance and therefore I would like to see this work accepted. I also think the paper could be improved a bit further by adding experiments that more closely validate the theoretical results (point 5), and by clarifying limitations/shortcomings to this work.

---

> ### Author Response · Authors · 2021-11-22
> **Response to Reviewer M2hT**
>
> We thank the reviewer for the very useful comments, and would like to address your concerns as follows:
>
> 1. Extending from GD to SGD is easy for the result that the linear model converges to some interpolator, because the expectation of the training error of the model converges to zero, which also implies the convergence of the model (See Appendix B.1). The difficult part is extending from GD to SGD for the approximation theorem, which till this point we do not know how it could be done. We will clearly state that we only consider GD in the new version, and we leave extension to SGD to future work.
>
> 2. "Each $D_i$ is a subset of $X \times Y$" is natural and it is not even an assumption. $X \times Y$ is the data domain, which means that all samples belong to $X \times Y$. And $D_i$ is a group (subdomain/environment) of samples, so of course it should be a subset of $X \times Y$. We would also like to remind the reviewer that we never assume that $D_i$ is known during training, and it has nothing to do with the model.
>
> 3. We will emphasize that we compare between reweighting algorithms and ERM that start from the same initial point in the new version.
>
> 4. The previous empirical observation is that downsampling majority groups by removing data from them to balance the group sizes can sometimes improve the worst-group performance. Previously people don't know why this could happen because intuitively having less data hurts the performance. Our results explain why this could be possible. Note that we do not prove that this "always" happens, which is also not true in practice. We will revise this paragraph to remove any confusion, but we will not remove this paragraph because we believe this is a very important takeaway from this paper.
>
> 5. This is a very good suggestion. In the new version we will include an experiment that uses very wide neural networks on synthetic datasets to empirically demonstrate our results.
>
> We hope that our response addresses your concerns and would like to answer any further questions.

---

### Official Review · Reviewer_wG4N · 2021-11-03

**Correctness:** 4
**Technical Novelty And Significance:** 2
**Empirical Novelty And Significance:** 1
**Recommendation:** 3
**Confidence:** 4

**Main Review:**

- The paper is studying an important problem about the effect of importance reweighing in overparameterized models which has attracted a lot of interest recently and thus, would be of interest to the community.
- However, the paper has many serious flaws. First of all, the paper is studying squared loss for its analysis and does not include a discussion about the choice of loss function. Loss function used in the previous works cited in this paper was cross entropy loss. So, there is this discrepancy between the works/experiments that this work is replicating and the theoretical results that are included.
 - This work is missing important citations for example [1] which initiated this study of importance reweighing methods in deep learning to the best of my knowledge.  Moreover, this has already been studied theoretically for cross entropy and other general losses by [2] which has not been cited in this work. So, it is not even clear if studying this paper for the squared loss brings out any new idea or insight on top of what was already known before. The point about regularization being necessary has already been made in previous works.

[1] What is the Effect of Importance Weighting in Deep Learning? Byrd et. al.
[2]  UNDERSTANDING THE ROLE OF IMPORTANCE WEIGHTING FOR DEEP LEARNING, Xu et al.

**Summary Of The Paper:**

This paper theoretically proves the empirical observation that has been made in a lot of recent works regarding the role of importance reweighing in overparameterized deep networks. For squared loss and independent data points, it theoretically proves that reweighing the data points has no effect on the final model learned for linear models and linearized neural networks. This work also analyzes the role of regularization in preventing this behavior and proves that regularization has to be large enough to prevent small training error to have any effect. This work also has supporting empirical results on two datasets- celeba and waterbirds.

**Summary Of The Review:**

- However, the paper has many serious flaws. First of all, the paper is studying squared loss for its analysis and does not include a discussion about the choice of loss function. Loss function used in the previous works cited in this paper was cross entropy loss. So, there is this discrepancy between the works/experiments that this work is replicating and the theoretical results that are included.
 - This work is missing important citations for example [1] which initiated this study of importance reweighing methods in deep learning to the best of my knowledge.  Moreover, this has already been studied theoretically for cross entropy and other general losses by [2] which has not been cited in this work. So, it is not even clear if studying this paper for the squared loss brings out any new idea or insight on top of what was already known before.

I would like to make recommendation for rejecting this paper.

---

> ### Author Response · Authors · 2021-11-22
> **Response to Reviewer wG4N**
>
> We thank the reviewer for the very useful comments, and would like to address your concerns as follows:
>
> 1. Regarding the loss function: In this work we consider the squared loss, which is different from previous work which considers the cross entropy loss. The reason is that with the cross entropy loss, the model cannot really converge to an interpolator: the norm of its weights will keep growing as the cross entropy loss goes to zero. Previous work [1] proved that for a linear model trained by minimizing the cross entropy loss with importance weighting, the direction of its weight will converge to a vector that does not depend on the sample weights, which is a negative result similar to our results. However, we cannot extend this result to sufficiently wide neural networks because the proof of the approximation theorem requires that the change in model weights is finite throughout training, which is not true for the cross entropy loss. We will add experiments that train models by minimizing the squared loss, and the citations the reviewer points out.
>
> 2. Regarding the novelty of this paper: We strongly disagree with the comment that this paper is the same as previous work and brings no new ideas, for the following three reasons:
>
> (a) To our best knowledge, all previous theoretical results, including the ones the reviewer points out, are limited to linear models and static reweighting algorithms (such as importance weighting). However, we prove our results for neural networks and dynamic reweighting algorithms, and the proof is non-trivial.
>
> (b) Maybe the reviewer believes that the transfer of results from linear models to sufficiently wide neural networks is simple with the previous NTK results, but it is in fact by no means simple. First, the previous famous (claimed) NTK result in [2] is wrong, and we fix their result and proofs, which we believe is already a great contribution; Second, the previous NTK result only considers ERM. Extending this result to static reweighting algorithms like importance weighting is indeed easy, but it is definitely not so for dynamic ones, let alone dynamic + regularization which we study in Section 4.
>
> (c) In this paper we do not prove that regularization is necessary. In fact, what we prove is a negative result. We prove that regularization does not work if it is not big enough to lower the training performance. To our best knowledge, this is a new theoretical result that does not appear in any previous work, though [3] did empirically show that large regularization is required, and our theoretical results justify their observations.
>
> Thus, we hope that the reviewer could reconsider the novelty and significance of this work. Though the results seem similar to previous work, we prove the results for a much more general setting. The extension is non-trivial, and the proof techniques in this work are useful for future work.
>
> We hope that our response addresses your concerns and would like to answer any further questions.
>
> [1] Da Xu, Yuting Ye, and Chuanwei Ruan. Understanding the role of importance weighting for deep learning. In International Conference on Learning Representations, 2021.
>
> [2] Lee et al., Wide neural networks of any depth evolve as linear models under gradient descent. Advances in neural information processing systems, 32:8572–8583, 2019.
>
> [3] Sagawa et al., Distributionally robust neural networks for group shifts: On the importance of regularization for worst-case generalization. In International Conference on Learning Representations, 2020.

---

### Official Review · Reviewer_hLzT · 2021-11-05

**Correctness:** 2
**Technical Novelty And Significance:** 3
**Empirical Novelty And Significance:** 2
**Recommendation:** 5
**Confidence:** 4

**Details Of Ethics Concerns:**

This is a paper about fairness through worst-group performance. Yet the abstract/intro are quite misleading and can have negative impact.

**Main Review:**

Understanding worst-group performance is an important and interesting topic. The paper is relatively easy to read, and provides valuable insights about this problem. However, there are some issues that need to be addressed

+ The abstract and introduction are quite misleading. For example, we see strong statements in the abstract such as "we cannot hope for reweighting algorithms to converge to a different interpolator than ERM..." or "we prove that for regularization to work, it must be large enough to prevent the model from achieving small training error." Such blanket statements without any hint that these are based on important assumptions (that are somewhat scattered in the paper) are misleading for the readers (of course unless if they dig deeper)

+ The first row of figure 1 is a good example of an overparametrized model, where the training accuracy of all models become 1 at some point in training. However, we still clearly see that even after than point, the test accuracies of different approaches are not the same. This is in contrast with the theoretical result by the authors that all of those methods would converge to the same result. The only explanation for this discrepancy is that the assumptions that the authors have made do not make sense for this experiment. For the second row (CelebA) the algorithms do not reach training error zero, and are effectively outside the regime of interest for the theoretical results of this paper. In conclusion, the the theoretical results do not really support the empirical results in this case; they really show different things.

+ Given the discrepancy between the theoretical and empirical results, it becomes more important for the authors to make the abstract/intro/.. more accurate. For example, someone that reads the abstract will conclude that there is no way for the reweighting to help unless the average accuracy drops significantly, whereas the experiments show otherwise.

+ The observation in section 3.1 that for linear models, the change in the parameters is always in the span of the data points is quite useful. I believe this is a standard observation that has been made before but I may be wrong. Can the authors comment about this? Are they the first to discover this? If not, they should at least add a citation.

+ More generally, are the other proofs (e.g, for the wide networks) similar to the analyses performed in previous work? If so, can you explicitly name those papers?

+ Is the theoretical analysis specific to the squared loss (e.g., in Thm 4, 5, 6)? If so, this must be clearly indicated as an extra assumption.

+ Can you elaborate on assumption 1? Why is it intuitively a reasonable assumption (aside from the empirical study)?

+ It looks like some of your experiments is similar to Sagawa et al. (2020a). Can you elaborate on the differences?

+ In Table 1, I believe you have left out the performance of ERM for early stopping. I think it is quite important to have those to be able to compare with the other methods.

+ As a side-comment, the practice of reporting 95% confidence intervals just based on running the method on a number of  random seeds is not necessarily valid. (usually you need to do cross-validation, or if not, argue that the test set is large enough...)

**Summary Of The Paper:**

The authors study the problem of improving the worst-group accuracy in the over-parametrized setting. They offer theoretical and empirical evidence that reweighting does not affect the final solution in this setting; in other words, the algorithm would converge to the same interpolators as that of basic ERM. The theoretical analysis is done for linear models, linearized networks, and wide fully connected networks.

The authors also argue that the solution does not change unless a considerable amount of regularization is incorporated, so much so that it affects the training error. Again, some theoretical and empirical evidence is provided.



**Summary Of The Review:**

The paper studies an important problem. The theoretical analysis is interesting though it has its own limitations. It will help if the authors comment about the novelty of the proof techniques compared to the previous work.

Many statements of the paper are misleading and should be addressed. The empirical study and the theoretical study do not necessarily support each other and this discrepancy has not been discussed in the paper.

---

> ### Author Response · Authors · 2021-11-22
> **Response to Reviewer hLzT**
>
> We thank the reviewer for the very detailed and helpful comments. We would like to address your concerns as follows:
>
> 1. There are three major points of novelty in our paper:
>
> (a) We prove for a large family of dynamic reweighting algorithms that a linear model converges to an interpolator, while to our best knowledge previous results are limited to ERM and static reweighting algorithms.
>
> (b) We prove the approximation theorem (Thm. 5) for dynamic reweighting algorithms, while the previous famous (claimed) result in [1] is limited to ERM, and their results and proofs are wrong, which we fix in this paper (See Appendix C).
>
> (c) We prove the approximation theorem for dynamic reweighting algorithms with regularization (Thm. 5), which also depends on new techniques.
>
> The observation that the change in parameters lies in a low-dimensional subspace is not new (e.g. see [2]), and we do not claim it to be new. We call it the "key observation" because it is useful and intuitive, and makes our results easier to understand. We will add the above citation in the new version.
>
> 2. We agree that the abstract and the introduction can be misleading, though in our theory part (Sections 3 and 4) we do clearly state the assumptions. We will rephrase the statements in the abstract and introduction, and state our assumptions in the introduction part.
>
> 3. Regarding Figure 1: For the CelebA dataset, the training accuracies do go to 1 when trained for sufficiently long time. Please see the new Figure 2(a) and 2(c). When wd = 0, the training accuracy goes to 1 for both importance weighting and group DRO. We will run the experiments for longer time (4000 epochs on Waterbirds and 1000 epochs on CelebA) which will make the conclusion more obvious, and update Figure 1 in the new version of the paper.
>
> 4. Regarding the discrepancy between theory and experiments: In our theory we prove for sufficiently wide feed-forward neural networks, while in the experiments the width is not sufficiently large, which causes the discrepancy between the theory and the experiments. Some may think that assuming the network to be sufficiently wide does not make sense, but this is the assumption most existing theory papers make, and the results derived from this assumption do provide people with lots of insights.
>
> 5. Regarding Assumption 1: As mentioned in the general response, we have weakened Assumption 1. It is necessary to assume that $q_i^{(t)}$ does not converge to 0 to prove that the model can fix all training samples, since otherwise we can always construct some adversarial $q_i^{(t)}$ which converges to 0 such that the model cannot fit sample $i$ (i.e. its loss on sample $i$ is at least $\epsilon$ for some $\epsilon$). "$q_i^{(t)}$ does not converge to 0" and the old Assumption 2 are equivalent to the new Assumption 1.
>
> 6. The results are limited to the squared loss. For other loss functions, as long as we can prove (1) the approximation theorem and (2) that the linear model converges to some interpolator, we can prove the same results. However, this means that we need to prove (1) and (2) for every loss function, so that's why we only focus on the squared loss. In the new version, we will clearly state in the introduction that our results only consider the squared loss, and discuss how to extend the result to other losses.
>
> 7. Sagawa et al. (2020a) only compares ERM to group DRO, and we also include importance weighting, which is more widely used.
>
> 8. We will add the experimental results for ERM + early stopping.
>
> We hope that our response addresses your concerns and would like to answer any further questions.
>
> [1] Lee et al., Wide neural networks of any depth evolve as linear models under gradient descent. Advances in neural information processing systems, 32:8572–8583, 2019.
>
> [2] Gunasekar et al., Characterizing implicit bias in terms of optimization geometry. In International Conference on Machine Learning, pages 1832–1841. PMLR, 2018.

---

> > ### Comment · Reviewer_hLzT · 2021-11-30
> > **read your comments**
> >
> > Thanks for making an effort to answer my questions. Some of my concerns are already addressed. But the authors mention that they will fix the other important concerns in the subsequent versions. I cannot update my score based on this promise but I think the authors will be able to improve the presentation in the later submissions.

---

### Author Response · Authors · 2021-11-22
**General Response to the Reviews**

We thank all the reviewers for their helpful comments.

We agree with a reviewer that while our abstract and introduction discussed general neural networks, our results focused on linear and NTK regime neural networks. This is in part because the community at large lacks tools for analyzing general neural networks. But we acknowledge the over-claiming in the introduction, and will adjust the manuscript accordingly.

But we note that even when restricted to linear and NTK-regime neural networks, showing that a large family of dynamic reweighting algorithms converge to an interpolator is not only novel (to the best of our knowledge previous results are limited to ERM and static reweighting algorithms), but was also considerably technically involved. In particular, we note that the famous NTK results and the proofs in [1] (which showed that a wide fully-connected neural network can be approximated by its linearized counterpart) was previously incorrect, and we spent considerable effort in fixing the proof and results, which we needed to for our extensions, as we note in the paper and Appendix C. We extended the (corrected) NTK result from ERM to not only dynamic reweighting algorithms, but also those that incorporate regularization.

We also note the following updates in the revised uploaded version of the paper:

1. In the experiments in Section 4.2, we train the models on the CelebA dataset for much more epochs than the original version (from 80 epochs to 250 epochs), and update Table 1, Figure 2 and the analysis in this section correspondingly. The overfitting phenomenon is not obvious when the models are only trained for 80 epochs. When trained for 250 epochs, the models clearly overfit, and it is more obvious that regularization needs to be large enough to lower the training performance.

2. We weaken Assumptions 1 and 2. In the new Section 3, we only assume that $q_i$ (which is the limit of $q_i^{(t)}$ as $t \rightarrow \infty$) is no less than some $q^* > 0$; and in the new Section 4 we make no assumption on the reweighting algorithms: the results hold for all static and dynamic reweighting algorithms.

3. We fix a small problem in the proof of Thm. 1.

[1] Lee et al., Wide neural networks of any depth evolve as linear models under gradient descent. Advances in neural information processing systems, 32:8572–8583, 2019.

---

### Decision · Program_Chairs · 2022-01-20

**Decision:**

Reject

**Comment:**

This paper studies the effect of importance weighting in three model classes: linear models, linearized networks, and wide fully connected networks, and show that under certain assumptions, gradient descent for training an overparameterized model converges to the same ERM interpolator regardless of the reweighting scheme. The reviewers acknowledge that this paper had good exposition and writing in general, but they were in consensus that the initial version of this paper includes many inaccurate overclaiming statements. In summary, after discussions, the reviewers would like the authors to:

- revise the abstract and the introduction, specifically, adding appropriate qualifiers on the neural networks, losses, full gradient descent training, etc (Reviewers hLzT, M2hT, fZgx)
- address the discrepancy between theory and experiments, e.g. the inconsistency of the loss chosen in theory and experiments, the requirement that the widths of the neural networks need to large (Reviewers wG4N, fZgx)
- add experimental results for early stopping (Reviewer hLzT)
- empirically verify that the final solutions of ERM, DRO, and IW initialized at the same \theta^0 are very similar (Reviewer M2hT)
- discuss the novelty compared to Sagawa et al in the paper (Reviewer wG4N)

thus, this submission needs a major revision, and is not ready for acceptance in its current form. We encourage the authors to revise accordingly, and resubmit in the future.